# Addressing Data Heterogeneity in Federated Learning with Adaptive Normalization-Free Feature Recalibration

## Abstract

Federated learning is a decentralized collaborative training paradigm preserving stakeholders' data ownership while improving performance and generalization. However, statistical heterogeneity among client datasets degrades system performance. To address this issue, we propose **Adaptive Normalization-free Feature Recalibration (ANFR)**, an architecture-level approach that combines weight standardization and channel attention. Weight standardization normalizes the weights of layers, making it less prone to mismatched client statistics and inconsistent averaging, ensuring robustness under heterogeneity. Channel attention produces learnable scaling factors for feature maps, suppressing inconsistencies across clients due to heterogeneity. We demonstrate that combining these techniques boosts model performance beyond their individual contributions, by improving class selectivity and channel attention weight distribution. ANFR works with any aggregation method, supports both global and personalized FL, and adds minimal overhead. When training with differential privacy, ANFR achieves an appealing balance between privacy and utility, enabling strong privacy guarantees without sacrificing performance. By integrating weight standardization and channel attention in the backbone model, ANFR offers a novel and versatile approach to the challenge of statistical heterogeneity. Extensive experiments show ANFR consistently outperforms established baselines across various aggregation methods, datasets, and heterogeneity conditions. Code is provided at
https://anonymous.4open.science/r/anfr_iclr_updated/.

## 1 Introduction

Federated learning (FL) (McMahan et al., 2017) is a decentralized training paradigm enabling clients to jointly develop a global model without sharing private data. By preserving data privacy and ownership, FL holds promise for applications in healthcare, finance, and mobile devices. A fundamental challenge in FL is statistically heterogeneous, i.e. non-independent and identically distributed (non-IID) client datasets, as they can degrade the performance of the global model and hinder convergence (Li et al., 2020b; Hsu et al., 2019). Addressing this is critical for FL's success in real-world scenarios.

Most prior research focuses on aggregation methods reducing client drift (Karimireddy et al., 2020) or tailoring model layers to client-specific data variations (Zhang et al., 2023). These strategies often overlook how model architecture affects performance under heterogeneity. Batch Normalization (BN) (Ioffe & Szegedy, 2015), effective in centralized settings, hinders performance in heterogeneous FL due to mismatched client-specific statistics and inconsistent parameter averaging (Wang et al., 2023; Guerraoui et al., 2024). In response, using other feature normalization methods like Group Normalization (GN) (Wu & He, 2018) and Layer Normalization (LN) (Ba et al., 2016) has been frequent (Hsieh et al., 2020; Reddi et al., 2021; Wang et al., 2021; Du et al., 2022). While avoiding mini-batch statistics, these alternatives often slow convergence and reduce performance compared to BN (Chen & Chao, 2021; Tenison et al., 2023; Zhong et al., 2024). Previous works have not designed models specifically tailored to combat heterogeneity, leaving a gap in FL research.

We address this gap in the image domain by proposing Adaptive Normalization-Free Feature Recalibration (ANFR), an architecture-level approach designed to enhance robustness in FL under

data heterogeneity. ANFR combines weight standardization (Qiao et al., 2020) with channel attention (Hu et al., 2018) to directly tackle the challenges posed by non-IID data. Weight standardization normalizes convolutional layer weights instead of activations, avoiding reliance on mini-batch statistics problematic in FL. This reduces susceptibility to mismatched statistics and inconsistent averaging. Channel attention generates learnable scaling factors for feature maps, suppressing features that are inconsistent across clients due to heterogeneity and emphasizing consistent ones. By integrating channel attention with weight-standardized models, ANFR enhances the model's ability to focus on shared, informative features across clients. This synergy boosts performance beyond the individual contributions of these components, enhancing class selectivity and optimizing channel attention weight distribution. ANFR works with any aggregation method and is effective in both global and personalized FL settings, with minimal computational overhead. Furthermore, when training with differential privacy, ANFR achieves an appealing balance between privacy and utility, enabling strong privacy guarantees without sacrificing performance.

We validate the effectiveness of ANFR through extensive experiments on a diverse set of datasets, including medical imaging and natural image classification tasks, under various types of data heterogeneity. Results demonstrate ANFR consistently outperforms established baselines across different aggregation methods, datasets, and heterogeneity conditions. By focusing on architectural components, our approach complements algorithmic advancements and addresses a crucial gap in FL research. The proposed model offers a robust and flexible solution to the challenge of statistical heterogeneity, contributing to the advancement of federated learning by enhancing performance, stability, and privacy-preserving capabilities.

## 2   RELATED WORK

Since McMahan et al. (2017) introduced FL, most research has focused on developing aggregation algorithms to address challenges like data heterogeneity. In global FL (GFL), methods such as proximal regularization (Li et al., 2020a) and cross-client variance reduction (Karimireddy et al., 2020) aim to reduce client drift. Techniques like discouraging dimensional collapse through correlation matrix norm regularization (Shi et al., 2023), adopting relaxed adversarial training (Zhu et al., 2023), and performing amplitude normalization in frequency space (Jiang et al., 2022) have also been proposed. Other recent ideas are constructing global pseudo-data to de-bias local classifiers and features (Guo et al., 2023), introducing concept drift-aware adaptive optimization (Panchal et al., 2023), and hyperbolic graph manifold regularizers (An et al., 2023). In personalized FL (pFL), personalizing layers of the model can mitigate heterogeneity. The simplest approach shares all model parameters except the classification head (Arivazhagan et al., 2019). More advanced methods replace lower layers and mix higher ones (Zhang et al., 2023) or adjust mixing ratios based on convergence rate approximations (Jiang et al., 2024). While these algorithmic approaches have advanced both GFL and pFL, they often overlook the impact of the underlying architecture on performance.

We address this gap by exploring how model components can enhance FL performance. This is orthogonal to algorithmic advancements, representing a crucially underdeveloped area. Previously, Qu et al. (2022) found using vision transformers instead of convolutional networks increased performance. Studies by Pieri et al. (2023) and Siomos et al. (2024) evaluated different architectures and aggregation methods, showing that changing the architecture, rather than the aggregation method, can be more beneficial. These works did not design models specifically tailored to combat heterogeneity. In contrast, our method integrates architectural components that enhance robustness across diverse client distributions into the model, directly addressing data heterogeneity.

The normalization layer has been a focal point of component examination as Batch Normalization (BN) (Ioffe & Szegedy, 2015) has been shown both theoretically (Li et al., 2021; Wang et al., 2023) and empirically (Hsieh et al., 2020; Du et al., 2022; Guerraoui et al., 2024) to negatively impact performance in heterogeneous FL. Mismatched local distributions lead to averaged batch statistics and parameters that fail to accurately represent any source distribution. The primary approaches addressing this issue are modifying the aggregation rule for the BN layer or replacing it entirely. Some methods keep BN parameters local (Li et al., 2021; Andreux et al., 2020) or stop sharing them after a certain round (Zhong et al., 2024). Others replace batch-specific statistics with shared running statistics when normalizing batch inputs to match local statistical parameters (Guerraoui et al., 2024) or leverage layer-wise aggregation to also match associated gradients (Wang et al.,

2023). These methods rely on decently sized batches to accurately approximate statistics and are incompatible with differential privacy. To replace BN, Group Normalization (GN) (Wu & He, 2018) has been frequently used (Hsieh et al., 2020; Reddi et al., 2021; Wang et al., 2021) since it does not rely on mini-batch statistics. However, tuning the number of groups in GN is required to maximize effectiveness and Du et al. (2022) showed that Layer Normalization (LN) (Ba et al., 2016) performs better than GN in some settings. Separate studies have shown both GN and LN offer inconsistent benefits over BN, depending on the characteristics and heterogeneity of the dataset (Tenison et al., 2023; Chen & Chao, 2021; Zhong et al., 2024).

We circumvent these issues by applying weight standardization (Qiao et al., 2020) to normalize the weights of the model instead of the activations. Inspired by Brock et al. (2021a), who showed that such Normalization-Free (NF) models can train stably and perform on par with BN in centralized learning, we explore this concept in FL. Previously, Zhuang & Lyu (2024) proposed an aggregation method specific to NF models for multi-domain FL with small batch sizes. Similarly, Siomos et al. (2024) showed that NF-ResNets improve upon vanilla ResNets under different initialization schemes and aggregation methods, while Kang et al. (2024) proposed a personalized aggregation scheme that replaces each BN layer with weight normalization (Salimans & Kingma, 2016) followed by a learnable combination of BN and GN. Additionally, our method adaptively recalibrates the resulting feature maps using channel attention modules, such as the Squeeze-and-Excitation block (Hu et al., 2018). By doing so, the model can focus more on relevant features across clients, effectively addressing data heterogeneity. Zheng et al. (2022) previously explored channel attention for pFL, proposing a modified channel attention block that is kept personal to each client. Unlike previous methods limited to specific aggregation strategies or settings, our approach can complement any heterogeneity-focused aggregation method, is effective even with large batch sizes, and supports various attention modules. Appendix C summarizes the differences between ANFR and related work. By integrating weight standardization with channel attention, ANFR provides a robust and flexible solution to data heterogeneity in FL, overcoming limitations of activation normalization techniques and complementing aggregation methods.

## 3 ADAPTIVE NORMALIZATION-FREE FEATURE RECALIBRATION

### 3.1 BACKGROUND AND NOTATION

We consider a FL setting with $C$ clients, each owning a dataset of image-label pairs $D_i = \{(x_k, y_k)\}$ and optimizing a local objective $\mathcal{L}_i(\theta) = \mathbb{E}_{(x,y)\sim D_i}[l(x,y;\theta)]$, where $l$ is a loss function and $\theta$ the model parameters. The global objective is learning a model $f(\theta)$ that minimizes the aggregate loss:

$$min_\theta \mathcal{L}(\theta) = \sum_{i=1}^{C} \frac{|D_i|}{|D|} \mathcal{L}_i(\theta) \tag{1}$$

Heterogeneity among $D_i$ can degrade the global model performance and slow convergence (Kairouz et al., 2021). In this study, we modify the backbone model to address this. As they are the most widely used family, and they perform better or on par with others (Pieri et al., 2023; Siomos et al., 2024), we focus specifically on convolutional neural networks (CNNs). Let $\mathbf{X} \in \mathbb{R}^{B \times C_{\text{in}} \times H \times W}$ represent a batch of $B$ image samples with $C_{\text{in}}$ channels and dimensions $H \times W$. For a convolutional layer with weights $\boldsymbol{W}$ and a kernel size of 1, the outputs are given by:

$$\mathbf{A} = \mathbf{X} * \boldsymbol{W} = \sum_{c=1}^{C_{\text{in}}} \boldsymbol{W}_{:,c} \mathbf{X}_{:,c,:,:} \, , \text{ where } \mathbf{A} \in \mathbb{R}^{B \times C_{\text{out}} \times H \times W}, \, \boldsymbol{W} \in \mathbb{R}^{C_{\text{out}} \times C_{\text{in}}} \tag{2}$$

In typical CNNs, the activations are then normalized:

$$\widehat{\mathbf{A}} = \frac{\boldsymbol{\gamma}}{\boldsymbol{\sigma_i}}(\mathbf{A}_i - \boldsymbol{\mu_i}) + \boldsymbol{\beta}, \quad \boldsymbol{\mu_i} = \frac{1}{|\mathbb{S}_i|} \sum_{k \in \mathbb{S}_i} \mathbf{A}_k, \quad \boldsymbol{\sigma_i^2} = \frac{1}{|\mathbb{S}_i|} \sum_{k \in \mathbb{S}_i} (\mathbf{A}_k - \mu_i)^2 \tag{3}$$

where $\boldsymbol{\beta}, \boldsymbol{\gamma} \in \mathbb{R}^{C_{\text{out}}}$ are learnable parameters, $\boldsymbol{i} = (i_N, i_C, i_H, i_W)$ is an indexing vector and $\mathbb{S}_i$ is the set of pixels over which $\boldsymbol{\mu_i}, \boldsymbol{\sigma_i}$ are computed. BN computes statistics along the $(B, H, W)$ axes, LN along $(C, H, W)$, and GN along $(C, H, W)$ separately for each of $\mathcal{G}$ groups of channels.

Channel attention (CA) mechanisms, like the Squeeze-and-Excitation (SE) block (Hu et al., 2018), recalibrate feature responses by modeling inter-channel relationships. The channel descriptor $\boldsymbol{Z} \in$

Figure 1: This example illustrates how channel attention can be useful in FL with data variability, by boosting $\mathcal{C}_R$ and suppressing $\mathcal{C}_{NR}$. Left: The two clients have heterogeneous datasets. Middle: An edge detector is robust to this feature shift; the activations are consistent for both clients. Right: A blue detector, however, is not robust and its activations cause conflicting gradients.

$\mathbb{R}^{B \times C_{\text{out}}}$ is obtained via Global Average Pooling (GAP):

$$\boldsymbol{Z} = (HW)^{-1} \sum_{h,w}^{H,W} \widehat{\mathbf{A}}_{:,:,h,w} \tag{4}$$

This descriptor is then non-linearly transformed to capture dependencies between channels; in SE blocks this is done via the learnable weights $\boldsymbol{W}_1 \in \mathbb{R}^{\frac{C_{\text{out}}}{r} \times C_{\text{out}}}$ and $\boldsymbol{W}_2 \in \mathbb{R}^{C_{\text{out}} \times \frac{C_{\text{out}}}{r}}$, where $r$ is a dimensionality reduction ratio:

$$\boldsymbol{S} = \sigma\left(\boldsymbol{W}_2 \delta\left(\boldsymbol{W}_1 \boldsymbol{Z}\right)\right), \quad \text{where} \quad \boldsymbol{S} \in \mathbb{R}^{B \times C_{\text{out}}}, \ \sigma : \text{sigmoid}, \ \delta : \text{ReLU} \tag{5}$$

yielding per-channel scaling factors $\boldsymbol{S}$ which are applied to the normalized activations $\tilde{\mathbf{A}} = \boldsymbol{S} \odot \widehat{\mathbf{A}}$.

### 3.2 EFFECT OF NORMALIZATION ON CHANNEL ATTENTION

In the presence of data heterogeneity, CA can suppress features sensitive to client-specific variations and emphasize consistent ones. In earlier layers, $\mathbf{A}$ consists of responses to filters detecting low-level features like colors and edges, while in later layers it contains class-specific features (Zeiler & Fergus, 2014). For the sake of explaining how CA impacts heterogeneous FL, we virtually partition filters into two distinct groups: those eliciting consistent features ($\mathcal{C}_R$) and inconsistent ones ($\mathcal{C}_{NR}$). Figure 1 illustrates an example. Both clients have images of airplanes and cars; Client 1's images have predominantly blue backgrounds, while Client 2's images have different backgrounds. Under this feature shift, edge-detecting filters produce consistent responses across both clients, thus belonging to $\mathcal{C}_R$, whereas filters sensitive to specific colors like blue activate differently across clients, forming $\mathcal{C}_{NR}$. While both activation types are informative locally, inconsistent activations from $\mathcal{C}_{NR}$ cause conflicting gradients during FL training. This motivates our use of CA in this context: during training, CA can assign higher weights to $\mathbf{A}_{\mathcal{C}_R}$ and lower weights to $\mathbf{A}_{\mathcal{C}_{NR}}$ without prior knowledge of which features belong to each set. The resulting adaptive recalibration aligns feature representations across clients, reducing gradient divergence and improving global model performance.

While CA mitigates the locality of convolution by accessing the entire input via pooling (Hu et al., 2018), if the normalization of $\widehat{\mathbf{A}}$ is ill-suited to heterogeneous FL, the input to (4) becomes distorted, leading to sub-optimal channel weights:

$$\boldsymbol{Z}^{\text{AN}} = \frac{\boldsymbol{\gamma}}{\boldsymbol{\sigma}_i HW} \sum_{h,w}^{H,W} \sum_{c=1}^{C_{\text{in}}} \boldsymbol{W}_{:,c} \mathbf{X}_{:,c,h,w} - \frac{\boldsymbol{\mu}_i \boldsymbol{\gamma}}{\boldsymbol{\sigma}_i} + \boldsymbol{\beta} \tag{6}$$

Activation normalization techniques suffer from this issue. BN is known to be problematic in heterogeneous settings for two reasons: mismatched client-specific statistical parameters lead to gradient divergence—separate from that caused by heterogeneity—between global and local models (Wang et al., 2023); and biased running statistics are used at inference (Guerraoui et al., 2024). Both contribute to well-established performance degradation (Li et al., 2021; Du et al., 2022). Since $\mu_i$ and $\sigma_i$ depend on batch-specific statistics, $\boldsymbol{Z}^{\text{AN}}$ varies across clients due to local distribution differences, leading to inconsistent channel descriptors, which in turn results in non-ideal channel weights. Aside from data heterogeneity, BN needs sufficient batch sizes to estimate statistics accurately, and is incompatible with differential privacy; these are limiting factors in resource-constrained and private FL scenarios. GN and LN also have drawbacks: GN normalizes within fixed channel groups, which may not align with the natural grouping of features, limiting its effectiveness under heterogeneity. LN assumes similar contributions from all channels (Ba et al., 2016), which is generally untrue for

CNNs, and clashes with our goal of reducing the influence of $\mathbf{A}_{\mathcal{C}_{NR}}$. Crucially, both normalize across channels to produce $\mu_i, \sigma_i$. This introduces additional channel inter-dependencies in (6), thus interfering with extracting representative channel descriptors.

### 3.3 ADAPTIVE NORMALIZATION-FREE FEATURE RECALIBRATION

To address these problems, we propose applying CA after normalizing the convolutional *weights* instead of the *activations* using Scaled Weight Standardization (SWS) from NF models (Brock et al., 2021a), which adds learnable affine parameters to weight standardization (Qiao et al., 2020):

$$\widehat{W}_{c_{\text{out}},c_{\text{in}}} = \frac{\gamma_{\text{eff},c_{\text{out}}}}{\sigma_{c_{\text{out}}}} \left( W_{c_{\text{out}},c_{\text{in}}} - \mu_{c_{\text{out}}} \right), \quad \mu_{c_{\text{out}}} = \frac{1}{C_{\text{in}}} \sum_{c=1}^{C_{\text{in}}} W_{c_{\text{out}},c}, \quad \sigma_{c_{\text{out}}}^2 = \frac{1}{C_{\text{in}}} \sum_{c=1}^{C_{\text{in}}} \left( W_{c_{\text{out}},c} - \mu_{c_{\text{out}}} \right)^2 \quad (7)$$

Here, $\gamma_{\text{eff}} = g \cdot \gamma / \sqrt{|C_{\text{in}}|}$ incorporates a learnable scale parameter $g$ and a fixed scalar $\gamma$ depending on the networks' non-linearity. We replace the normalized activation $\widehat{\mathbf{A}}$ with $\mathbf{A}' = \mathbf{X} * \widehat{W} + \boldsymbol{\beta}$. From (7) we observe that SWS does not introduce a mean shift ($\mathbb{E}[\mathbf{A}'] = \mathbb{E}[\widehat{\mathbf{A}}] = 0$), and preserves variance ($\text{Var}(\mathbf{A}') = \text{Var}(\mathbf{A})$) for the appropriate choice of $\gamma$, allowing stable training. By replacing normal convolutions with the ones described by (7), and following the signal propagation steps described in Brock et al. (2021a), we can train stable CNNs without activation normalization. We term this combination of weight standardization and channel attention Adaptive Normalization-Free feature Recalibration (ANFR). The input to (4) when using ANFR is:

$$Z^{\text{ANFR}} = \frac{\boldsymbol{\gamma}_{\text{eff}}}{\boldsymbol{\sigma} HW} \sum_{h,w}^{H,W} \sum_{c=1}^{C_{\text{in}}} \boldsymbol{W}_{:,c} \mathbf{X}_{:,c,h,w} - \frac{\boldsymbol{\mu}\boldsymbol{\gamma}_{\text{eff}}}{\boldsymbol{\sigma} HW} \sum_{h,w}^{H,W} \sum_{c=1}^{C_{\text{in}}} \mathbf{X}_{:,c,h,w} + \boldsymbol{\beta} \quad (8)$$

Comparing (6) and (8), we note several advantages of ANFR. First, $\boldsymbol{\sigma}$ and $\boldsymbol{\mu}$ are computed from convolutional weights, not the activations. Since weights are initialized identically and synchronized during FL, these weight-derived statistics are consistent across clients. Moreover, the second term of (8) now captures statistics of the input *before* convolution, providing an additional calibration point for CA and bypassing the effect of $\mathcal{C}_{NR}$. By applying CA after SWS, we ensure channel descriptors are not distorted by batch-dependent statistics or cross-channel dependencies introduced by activation normalization. This allows CA to adjust channel responses effectively, improving the model's capacity to learn stable feature representations that are consistent across clients with diverse data distributions. Therefore, the combination of SWS and CA overcomes the drawbacks of traditional normalization methods in federated learning, providing a novel and effective solution for improving model performance in the presence of data variability.

### 3.4 MECHANISTIC INTERPRETABILITY ANALYSIS

Next, we conduct a mechanistic interpretability analysis comparing the effects of BN and SWS on class selectivity and attention weight variability to further substantiate the effectiveness of integrating CA with SWS. We examine how well the ANFR model discriminates between classes before[1] and after training on the heterogeneous 'split-3' partitioning of CIFAR-10 from Qu et al. (2022). This evaluation helps understand how our method improves class discriminability under data heterogeneity. We isolate the effect of different components by comparing ANFR (using SWS with CA), BN-ResNet (using BN), NF-ResNet (using SWS without CA), and SE-ResNet (using CA with BN). Class selectivity is quantified by the class selectivity index (CSI) (Morcos et al., 2018), defined for each neuron as $\text{CSI} = (\mu_{\text{max}} - \mu_{-\text{max}})/(\mu_{\text{max}} + \mu_{-\text{max}})$, where $\mu_{\text{max}}$ is the class-conditional activation that elicits the highest response and $\mu_{-\text{max}}$ is the mean activation for all other classes. A right-skewed CSI distribution indicates higher class selectivity, crucial for effective classification under heterogeneous data. Lastly, we examine the distribution of attention weights, like done in Wang et al. (2020), for models using CA, to understand its contribution to class discrimination.

Figure 2 shows CSI distributions for the last layer before the classifier, where class specificity is maximized in CNNs. Before FL training, incorporating CA in SE-ResNet slightly increases class selectivity compared to BN-ResNet. Combining CA with SWS in ANFR shows negligible change

---

[1]All networks are pre-trained on ImageNet.

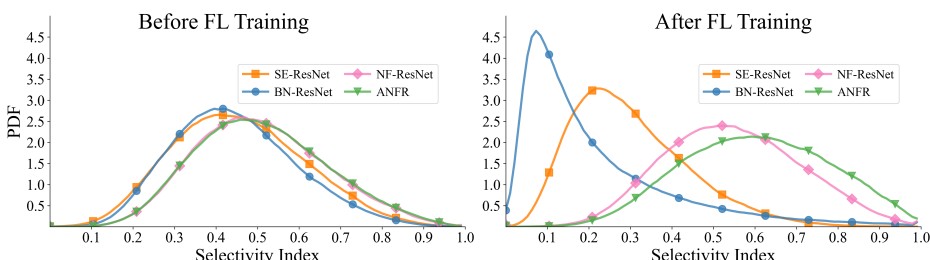

Figure 2: Left: CSI distributions before FL training, queried after the last CA module. Both normalizations (BN and SWS) show similar behavior, and CA has a minor impact. Right: after FL training, CA increases class selectivity, especially in conjunction with SWS in ANFR.

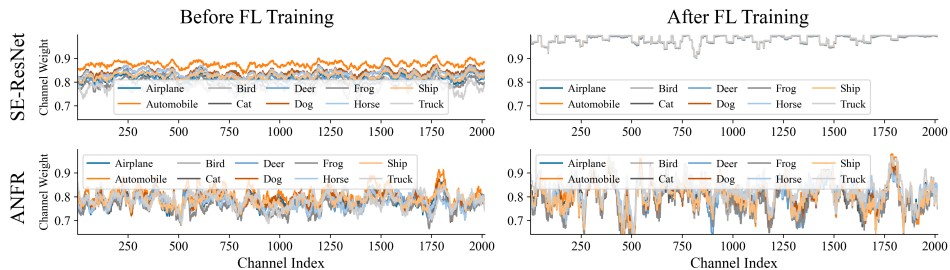

Figure 3: Top: Weights of the last CA module for SE-ResNet-50. Bottom: Same for ANFR-50. Left: Before FL training, CA provides a diverse signal varying across classes and channel indices to both models. Right: After FL training, the CA module in SE-ResNet degenerates to an identity. In ANFR, CA shows increased variability as it works to combat heterogeneity.

in class selectivity compared to NF-ResNet, indicating CA' minimal impact at this stage. However, after training on heterogeneous data, we observe a notable shift: BN reduces class selectivity (compared to before training), evidenced by left-skewed distributions for BN-ResNet and SE-ResNet. Adding CA increases class selectivity for both normalization methods, but due to receiving inconsistently normalized inputs (6) cannot fully mitigate BN's negative effect. The ANFR model, however, shows a significant increase in class selectivity compared to NF-ResNet, with strong class selectivity (CSI>0.75) units nearly doubling from ∼11% to ∼21%. This improvement manifests only after FL training, indicating that combining CA and SWS in ANFR enhances the model's ability to specialize and discriminate classes under data heterogeneity.

In Figure 3 we use the variability of attention weights across channels and classes as an indicator of adaptation: high variability suggests CA is actively re-weighing features to adapt to different class characteristics. Before FL training (left panel), both SE and ANFR models display high variability, as, when heterogeneity is not a factor, CA provides a diverse and informative signal for both activation and weight normalization. After FL training (right panel), the attention mechanism of SE-ResNet turns into an identity operator, with attention weights converging to 1 across all channels and classes, meaning SE-ResNet fails to preserve the discriminative power of CA under heterogeneity. In contrast, ANFR maintains high variability in CA weights across channels and classes. This sustained variability implies that CA remains active and continues to provide class-discriminative signals when combined with weight standardization.

These insights support our design choices. BN's adverse effects in heterogeneous FL are highlighted by diminished class selectivity and inactive CA in SE-ResNet, while ANFR maintains and improves class selectivity, demonstrating that integrating CA with weight standardization effectively counters data heterogeneity. The enhanced class selectivity in ANFR correlates with improved downstream performance in heterogeneous FL settings, as we show in Section 4. Additional details and extended CSI and attention weight results from other layers are presented in Appendix E.

## 4 EXPERIMENTS

### 4.1 EXPERIMENTAL SETTINGS

**Datasets.** We evaluate our approach on five classification datasets, including Fed-ISIC2019 (Ogier du Terrail et al., 2022) containing dermoscopy images from 6 centers with 8 classes where label distribution skew and heavy quantity skew is present; FedChest, a novel chest X-Ray multi-label dataset with 4 clients and 8 labels with label distribution skew and covariate shift; a partitioning of CIFAR-10 (Krizhevsky et al., 2009) which simulates heavy label distribution skew across 5 clients using the Kolmogorov-Smirnov (KS) 'split-2' as presented in Qu et al. (2022); CelebA (Liu et al., 2015) from the LEAF suite (Caldas et al., 2018), a binary classification task in a cross-device setting with a large number of clients, covariate shift and high quantity skew; and FedPathology, a colorectal cancer pathology slide dataset with 9 classes derived from Kather et al. (2019), featuring challenging concept drift as the images, which we do not color-normalize, were produced using two different staining protocols. FedChest contains images from PadChest (Bustos et al., 2020), CXR-14 (Wang et al., 2017) and CheXpert Irvin et al. (2019), which present one or more of 8 common disease labels. For FedPathology, used for DP training in Section 4.3, Dirichlet distribution sampling (Hsu et al., 2019) with $\alpha=0.5$ is used to simulate a moderate label distribution skew and partition the data to 3 clients. Each task covers a different aspect of the multi-faceted problem of data heterogeneity in FL, including different domains and sources of heterogeneity, to provide a robust test bed. More details are presented in Appendix A.1, including instructions to replicate FedChest in D.1.

**Compared models.** We compare ANFR with a typical ResNet (utilizing BN), a ResNet where BN is replaced by GN, a SE-ResNet (Hu et al., 2018), and a NF-ResNet. This selection isolates the effects of our architectural changes compared to using BN, using its popular substitution GN, and using weight standardization and CA separately. We choose a depth of 50 layers for all models to balance performance with computational expense. All models used in Section 4 are pre-trained on ImageNet (Russakovsky et al., 2015) using `timm` (Wightman, 2019), but additional experiments with randomly initialized models are presented in Appendix B.2. ANFR follows the structure of NF-ResNet, with the addition of CA blocks in the same position as SE-ResNet. Except for Section 4.4, we employ Squeeze-and-Excitation (Hu et al., 2018) as the attention mechanism. Additional model and computational overhead details are provided in Appendix A.3.

**Evaluated methods.** We use 4 global FL (GFL) and 2 personalized FL (pFL) aggregation methods as axes of comparison for the models, each representing a different approach to model aggregation: the seminal **FedAvg** (McMahan et al., 2017) algorithm, **FedProx** (Li et al., 2020a), which adds a proximal loss term to mitigate drift between local and global weights, **SCAFFOLD** (Karimireddy et al., 2020), which corrects client drift by using control variates to steer local updates towards the global model, **FedAdam** (Reddi et al., 2021), which decouples server-side and client-side optimization and employs the Adam optimizer (Kingma & Ba, 2017) at the server for model aggregation, **FedBN** (Li et al., 2021) which accommodates data heterogeneity by allowing clients to maintain their personal batch statistics, and by construction is only applicable to models with BN layers, and **FedPer** (Arivazhagan et al., 2019) which personalizes the FL process by keeping the weights of the classifier head private to each client. We note our proposal is an architectural one which is aggregation method-agnostic, thus we selected these widely known aggregation methods to represent a spectrum of strategies, from standard averaging to methods addressing client drift and personalization. This provides a robust comparison concentrated on the model architectures.

**Evaluation metrics.** For Fed-ISIC2019, we report the average balanced accuracy due to heavy class-imbalance as in (Ogier du Terrail et al., 2022). For FedChest, a multi-label classification task with imbalanced classes, we report the mean AUROC on the held-out test in this section and more metrics in Appendix D.2. We report the average accuracy for the other 3 datasets. In pFL settings, the objective is providing good in-federation models so we report the average metrics of the best local models, as suggested in (Zhang et al., 2023).

**Implementation Details.** We select hyper-parameters for each dataset by tuning the BN-ResNet (using the ranges detailed in Appendix A.2) and then using the same parameters for all models. This means the results in Section 4.2 are a conservative floor of the improvements that can be achieved, and in Appendix B.3 we show tuning for ANFR can further increase improvements. In Fed-ISIC2019 clients use Adam with a learning rate of 5e-4 and a batch size of 64 to train for 80 rounds of 200 steps. This setup is distinct from the one used in Ogier du Terrail et al. (2022) resulting

in performance improvements for all models. In Appendix B.1 we provide additional results using the original settings. In FedChest clients use Adam with a learning rate of 5e-4 and a batch size of 128 to train for 20 rounds of 200 steps. For DP-training in FedPathology, we set the probability of information leakage $\delta$ to $0.1/|D_i|$, as is common, the noise multiplier to 1.1, the gradient max norm to 1.0, and train for 25 rounds, which is the point where the models have expended a privacy budget of $\varepsilon=1$. For CelebA and CIFAR-10 we follow the settings of Qu et al. (2022); Pieri et al. (2023) which were tuned by the authors. All experiments are run in a simulated FL environment with NVFLARE (Roth et al., 2022) and PyTorch (Paszke et al., 2019), using 2 NVIDIA A100 GPUs for training. We report the mean and standard deviation across 3 seeds.

## 4.2 PERFORMANCE ANALYSIS AND COMPARISON

Table 1: Performance comparison across all architectures under different global FL aggregation methods and different datasets. Best in bold, second best underlined. ANFR consistently outperforms the baselines, often by a wide margin.

| Dataset | Method | Architecture | | | | |
|---|---|---|---|---|---|---|
| | | BN-ResNet | GN-ResNet | SE-ResNet | NF-ResNet | ANFR (Ours) |
| Fed-ISIC2019 | FedAvg | 66.01±0.73 | 65.09±0.42 | 65.29±1.32 | 72.49±0.60 | **74.78±0.16** |
| | FedProx | 66.49±0.41 | 66.51±1.21 | 66.29±0.63 | 71.28±2.14 | **75.61±0.71** |
| | FedAdam | 65.88±0.67 | 64.60±0.39 | 65.18±1.90 | 69.96±0.14 | **73.02±0.93** |
| | SCAFFOLD | 65.41±0.72 | 68.84±0.46 | 68.99±0.18 | 73.30±0.50 | **76.52±0.60** |
| FedChest | FedAvg | 82.80±0.13 | 83.40±0.25 | 82.14±0.18 | 83.40±0.11 | **83.49±0.14** |
| | FedProx | **82.14±0.10** | 82.04±0.08 | 81.50±0.26 | 81.26±0.58 | **82.14±0.10** |
| | FedAdam | 83.02±0.11 | 82.11±0.10 | 82.72±0.16 | 83.10±0.09 | **83.33±0.07** |
| | SCAFFOLD | 83.52±0.14 | 83.95±0.05 | 83.50±0.08 | 84.06±0.02 | **84.26±0.10** |
| CIFAR-10 | FedAvg | 91.71±0.74 | 96.60±0.11 | 94.07±0.04 | 96.72±0.05 | **97.42±0.01** |
| | FedProx | 95.03±0.04 | 96.05±0.04 | 94.60±0.07 | **96.82±0.04** | 96.33±0.09 |
| | FedAdam | 91.23±0.29 | 95.80±0.24 | 94.09±0.17 | 95.54±0.10 | **96.93±0.06** |
| | SCAFFOLD | 92.51±0.99 | 96.78±0.01 | 94.30±0.03 | 96.84±0.01 | **97.38±0.03** |

Table 2: pFL aggregation method comparison on Fed-ISIC2019 and FedChest. FedBN is only applicable to models using BN layers. ANFR remains the top performer.

| Dataset | Method | Architecture | | | | |
|---|---|---|---|---|---|---|
| | | BN-ResNet | GN-ResNet | SE-ResNet | NF-ResNet | ANFR (Ours) |
| Fed-ISIC2019 | FedPer | 82.36±0.80 | 80.66±0.47 | 81.22±0.77 | 84.2±0.43 | **84.94±0.46** |
| | FedBN | 82.82±0.06 | N/A | 81.84±0.28 | N/A | N/A |
| FedChest | FedPer | 83.39±0.10 | 83.73±0.10 | 83.36±0.14 | 83.70±0.14 | **83.8±0.14** |
| | FedBN | 83.38±0.12 | N/A | 83.33±0.14 | N/A | N/A |

**GFL scenario.** Average results for all datasets, models, and GFL aggregation methods are presented in table 1. First, we observe that GN does not consistently outperform the vanilla ResNet, supporting our pursuit of a more reliable alternative. For instance, GN is outperformed by BN in half of the tested aggregation methods on Fed-ISIC2019 and FedChest. Second, the sub-optimality of CA operating on BN-normalized features is evident, as the SE model frequently performs worse than BN-ResNet, notably across all aggregation methods on FedChest. NF-ResNet shows strong performance across all tasks and methods, confirming the potential of replacing activation normalization with weight standardization in FL. However, our proposed ANFR model consistently outperforms NF-ResNet, often by a considerable margin. For example, on Fed-ISIC2019 with SCAFFOLD, ANFR surpasses NF-ResNet's mean balanced accuracy by more than 3%. For the FedChest dataset, we employ a large batch size of 128 to maximize the probability that all classes are represented in each batch, following best practices for multi-label, class-imbalanced datasets. This is further analyzed in a batch size ablation in Appendix D.3. ANFR emerges as the top-performing model across aggregation methods and our results indicate that integrating CA with SWS networks provides sig-

nificant performance gains, suggesting that channel attention is a crucial component in designing effective FL models.

**pFL scenario.** Table 2 presents the results for pFL scenarios on Fed-ISIC2019 and FedChest. In FedChest, where images are grayscale and we use a large batch size, FedBN and FedPer are virtually equal: BN-ResNet achieves an AUROC of 83.38% with FedBN and 83.39% with FedPer, indicating that the estimated BN statistics closely match the true ones. GN-ResNet attains 83.73% with FedPer, slightly outperforming BN-ResNet, but ANFR with FedPer is the most performant option across both aggregation methods, yielding a mean AUROC of 83.8%. Conversely, under the severe label and quantity skew on Fed-ISIC2019, employing FedBN improves performance over FedPer for models employing BN. ANFR achieves the highest balanced accuracy of 84.94% nonetheless. Notably, GN performs worse than BN on Fed-ISIC2019, and the ineffectiveness of combining BN and CA is further evidenced, as SE-ResNet is outperformed by BN-ResNet in all scenarios. These findings demonstrate that adopting ANFR enhances performance across both datasets, leading to the best overall models. Unlike the trade-offs observed with BN-FedBN and GN-FedPer combinations, ANFR consistently outperforms other architectures across varying levels of data heterogeneity.

**Cross-device experiments on CelebA.** Table 3 presents the results of our models on the cross-device setting of CelebA, which contains 200,288 samples across 9,343 clients. While the binary classification task is relatively straightforward for individual clients, it poses challenges at the server level due to the vast number of clients and significant quantity and class skews—some clients have only a few samples or labels from a single class. We observe that ANFR outperforms the baseline models, demonstrating its adaptability across diverse FL scenarios.

Table 3: Performance Comparison in a cross-device setting, training with FedAvg on CelebA. The training setup follows Pieri et al. (2023), where 10 clients participate at each round until all clients have trained for 30 rounds. ANFR outperforms the baselines.

| Architecture | BN-ResNet | GN-ResNet | SE-ResNet | NF-ResNet | ANFR (Ours) |
|---|---|---|---|---|---|
| Average Accuracy | 82.2±1.21 | 85.41±0.68 | 85.55±0.84 | 88.17±0.3 | **88.91±0.28** |

## 4.3 SAMPLE-LEVEL DIFFERENTIALLY PRIVATE TRAINING

In privacy-preserving scenarios involving differential privacy (DP), BN cannot be used as calculating mini-batch statistics violates privacy-preservation so it is customarily replaced by GN. We demonstrate the utility of ANFR in such settings using the FedPathology setup described in Section 4.1. We train using DP-SGD with strict sample-level privacy guarantees: following good practices, the probability of information leakage $\delta$ is set to $0.1/|D_i|$, the noise multiplier is set to 1.1 and the gradient max norm to 1. We employ a privacy budget of $\varepsilon{=}1$, followed by training without privacy constraints ($\varepsilon{=}\infty$), to illustrate the privacy/utility trade-off of each model.

From the results presented in Table 4, we observe that with an unrestricted privacy budget, GN and ANFR perform comparably. However, when a strict budget is enforced GN suffers a sharp performance decrease of 17%, as expected following previous research (Klause et al., 2022), whereas ANFR's average accuracy is reduced by only 3%. ANFR's robustness under DP may be attributed to its reliance on weight standardization, which has been shown to benefit from additional regularization (Brock et al., 2021b; Zhuang & Lyu, 2024) such as that provided by DP-SGD's gradient clipping and gradient noising. Our experiments show DP training induces a regularization ef-

Table 4: Accuracy on the validation set of FedPathology when training with and without DP. Performance degrades severely for GN, while ANFR retains good performance.

| Privacy Budget | $\varepsilon = \infty$ | $\varepsilon = 1$ |
|---|---|---|
| GN-ResNet | **84.79±2.72** | 67.27±5.08 |
| ANFR (Ours) | 84.47±3.08 | **81.11±0.33** |

fect that disproportionately benefits NF models like ANFR, an observation also reported by De et al. (2022). These findings make ANFR a promising candidate for furthering development and adoption of DP training in FL, thereby enhancing the privacy of source data contributors, such as patients.

## 4.4 ATTENTION MECHANISM COMPARISON

Next, we investigate the impact of different attention mechanisms on performance. We compare the SE module used in previous sections with ECA (Wang et al., 2020), and CBAM (Woo et al., 2018). ECA replaces SE's fully-connected layers with a more efficient 1-D convolution to capture local cross-channel interactions. CBAM combines channel and spatial attention and utilizes both max and average pooling to extract channel representations. From Table 5 we observe that even the lowest-performing module on each dataset outperforms all baseline models from Tables 1 and 3, proving the robustness of our approach. No single mechanism consistently performs best, making further exploration of attention module designs an interesting avenue for future work.

Table 5: Comparing different channel attention modules after FL training with FedAvg. No module is consistently the best, but even the worst outperforms the best baseline (NF-ResNet).

|            | CIFAR-10          | Fed-ISIC2019      | FedChest          | CelebA            |
|------------|-------------------|-------------------|-------------------|-------------------|
| NF-ResNet  | $96.72 \pm 0.05$  | $72.49 \pm 0.60$  | $83.40 \pm 0.11$  | $88.17 \pm 0.30$  |
| ANFR (SE)  | $\mathbf{97.42 \pm 0.01}$ | $74.78 \pm 0.16$ | $83.49 \pm 0.14$ | $88.91 \pm 0.28$ |
| ANFR (ECA) | $97.13 \pm 0.11$  | $\mathbf{75.07 \pm 0.48}$ | $\mathbf{83.62 \pm 0.10}$ | $89.07 \pm 0.43$ |
| ANFR (CBAM)| $97.05 \pm 0.08$  | $74.19 \pm 0.68$  | $83.47 \pm 0.15$  | $\mathbf{89.31 \pm 0.41}$ |

## 4.5 QUALITATIVE LOCALIZATION PERFORMANCE COMPARISON

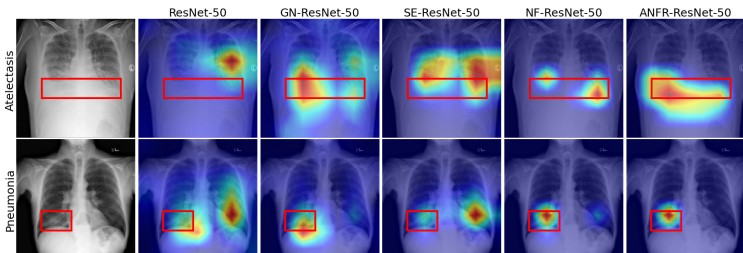

Figure 4: Comparison of the saliency maps generated by Grad-CAM++ from different architectures for a Pneumonia and an Atelectasis image, overlaid with ground-truth bounding boxes. We note ANFR improves localization and reduces activations outside the area of interest.

Finally, we assess the localization capability of each architecture after FL training with the best aggregation method on FedChest, SCAFFOLD. We compare the bounding box annotations provided by Wang et al. (2017) with Grad-CAM++ (Chattopadhay et al., 2018) heatmaps generated for samples labeled *Atelectasis* or *Pneumonia* from the FedChest test set. Figure 4 shows that ANFR's heatmaps more closely align with the annotated bounding boxes. This improved localization aids model interpretability, which is crucial in areas like medical imaging.

## 5 CONCLUSION

We introduce ANFR, a simple approach combining the strengths of weight standardization and channel attention to address the challenges of data heterogeneity and privacy at a design level in FL. ANFR fills a gap by being the first method to simultaneously work in GFL, pFL, and private FL scenarios while being compatible with any aggregation method and offering a robust increase in performance. Extensive experiments demonstrate the superior adaptability and performance of ANFR, as it consistently surpasses the performance of baseline architectures, regardless of the aggregation method employed. Our results position ANFR as a compelling backbone model suitable for both global and personalized FL scenarios where statistical heterogeneity and privacy guarantees are important concerns. Our findings highlight the need to look beyond aggregation methods as the core component of federated performance and the critical role of architectural innovations in reaching the next frontier in private and collaborative settings.

REPRODUCIBILITY STATEMENT

We have taken several steps to ensure reproducibility of our results. Details for the datasets we used and their pre-processing are in Appendix A.1. We also introduce a dataset of our own creation, Fed-Chest, and provide detailed instructions on how to reproduce it in Appendix D. How we tuned our models is explained in Appendix A.2. This paper introduces a model, so we will provide ImageNet pre-trained weights for it upon acceptance. Details for all the models we use are in Appendix A.3. Our code is available for inspection at `https://anonymous.4open.science/r/anfr_iclr_updated/` and we will properly open-source all of our code upon acceptance.

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

# A ADDITIONAL IMPLEMENTATION DETAILS

## A.1 DATASETS

**Skin Lesion Classification on Fed-ISIC2019.** Fed-ISIC2019 (Ogier du Terrail et al., 2022) contains 23,247 dermoscopy images from 6 centers across 8 classes and is a subset of the ISIC 2019 challenge dataset. We follow the original pre-processing, augmentation, loss, and evaluation metric of (Ogier du Terrail et al., 2022). This means the loss function is focal loss weighted by the local class percentages at each client, and the reported metric is balanced accuracy, as countermeasures against class imbalance. The augmentations used include random scaling, rotation, brightness changes, horizontal flips, shearing, random cropping to $200 \times 200$ and Cutout (DeVries, 2017). We train for 80 rounds of 200 local steps with a batch size of 64. The clients locally use Adam (Kingma & Ba, 2017), a learning rate of 5e-4, and a cyclical learning rate scheduler (Smith, 2017). In terms of heterogeneity, Fed-ISIC2019 represents a difficult task due to class imbalance and heavy dataset size imbalance, with the biggest client owning more than 50% of the data and the smallest client 3%.

**CIFAR-10**. Krizhevsky et al. (2009) consists of 50,000 training and 10,000 testing $32 \times 32$ images from 10 classes. We follow the setup of Pieri et al. (2023), specifically the 'split-2' partitioning where each client has access to four classes and does not receive samples from the remaining six classes. This means we train for 100 rounds of 1 local epoch with a batch size of 32. Clients use SGD with a learning rate of 0.03 and a cosine decay scheduler, in addition to gradient clipping to 1.0. During training the images are randomly cropped with the crop size ranging from 5% to 100% and are then resized to $224 \times 224$.

**CelebA from LEAF.** A partitioning of the original CelebA (Liu et al., 2015) dataset by the celebrity in the picture, this dataset contains 200,288 samples across 9,343 clients. The task is binary classification (smiling vs not smiling). We follow the setup presented in Pieri et al. (2023), training with 10 clients each round until all clients have trained for at least 30 rounds. The other settings are the same as those for CIFAR-10.

**FedPathology Slide Classification Dataset.** A colorectal cancer pathology slide dataset (Kather et al., 2019), consisting of $100k$ training images of Whole Slide Image (WSI) patches with labels split among 9 classes, is used to simulate a federation of 3 clients. We mimic one of the most important challenges in the WSI field by not color-normalizing the images, which come from two different labs with differences in staining protocols. The original $7k$ color-normalized validation set from Kather et al. (2019) is kept as a common validation set. We follow common practice (Hsu et al., 2019) to simulate label skew data heterogeneity by using a Dirichlet distribution with $\alpha = 0.5$ to partition the data. Since this artificial partitioning is random, we make sure to use the same seeds across architectures and privacy settings to compare on exactly the same partitioning instances. Our pipeline is built using Opacus (Yousefpour et al., 2022) and $(\alpha, \delta)$-Renyi Differential Privacy (RDP) (Mironov, 2017). Following good practices, the probability of information leakage $\delta$ is set to $0.1/|D_i|$ where $|D_i|$ represents each client's dataset size. The DP-specific hyper-parameters of the noise multiplier and gradient max norm are set to 1.1 and 1, respectively. Data augmentation includes random horizontal and vertical flips, random color jittering, and random pixel erasing. Clients use Adam with a learning rate of 5e-5, training for 500 local steps with a batch size of 64. Federated training is stopped after 25 rounds, which is the point where both architectures have expended, on average, a privacy budget of $\varepsilon = 1$. Finally, we train without using DP under the same settings to form a clearer picture of the privacy/utility trade-off of each model.

**Chest X-Ray Multi-Label Classification on FedChest.** Please refer to Appendix D.1.

## A.2 HYPER-PARAMETER TUNING

Hyper-parameters were optimized for the BN-ResNet and then the same parameters were used for all networks. The ranges were as follows:

- **Local Steps**: $\{100, 200, 500\}$
- **Rounds**: $\{20, 50, 75, 100\}$
- **Batch size**: $\{32, 64, 128\}$

- **Gradient Clipping**: {None, Norm Clipping to 1, Adaptive Gradient Clipping (Brock et al., 2021b)}

- **Learning rate**: {5e-5 − 1e-2}

- **Optimizer**: {Adam, AdamW, SGD with momentum}

- **Scheduler**: {None, OneCycleLR, Cosine Annealing, Cosine Annealing with Warm-up}

- **FedProx** $\mu$: {1e-3, 1e-2, 1e-, 2}

- **FedAdam Server learning rate**: {5e-4, 1e-3, 1e-2, 1e-1}

**Discussion.** We found both FL aggregation methods that introduce hyper-parameters difficult to tune: FedProx Li et al. (2020a) made a negligible difference for small $\mu$ values and decreased performance as we increased it; the server learning rate in FedOpt has to be chosen carefully, as large (1e-2,1e-1) learning rates led to non-convergence and small ones to disappointing performance. Gradient clipping helped ANFR but was detrimental to the vanilla ResNet. We found the use of a scheduler to be very beneficial for performance, as well as making the optimizer and initial learning rate choice less impactful. We store the intermediate learning rate at each client between rounds and resume the scheduler, and also follow this for the momentum buffers of the adaptive optimizers.

## A.3 MODEL DETAILS AND COMPUTATIONAL OVERHEAD

Table 6 presents pre-training details, parameter counts, multiply-accumulate counts (GMACs) and floating point operation counts (FLOPs) and ImageNet (Russakovsky et al., 2015) validation set top-1 performance for all models. For models which are pre-trained by us, links to the pre-trained weights will be made public after acceptance. Additionally, to gauge the computational overhead of ANFR, and by extension its applicability in low-resource environments, we compare training times for BN-ResNet-26 with those for ANFR-26 using ECA as the attention mechanism. The batch size is set to 32, and we measure the average time per iteration of forward + backward pass across 100 iterations using PyTorch's profiler. We do this for two distinct scenarios: devices without a CUDA-enabled GPU (e.g., smartphones), and devices with CUDA-enabled GPUs (e.g., edge devices such as Nvidia Jetson). The results in Table 7 show ANFR introduces marginal overhead ($\sim$10% without CUDA, $\sim$10% with CUDA) while providing a significant performance advantage, showcasing its practicality in resource-constrained settings.

Table 6: Comparison of model details. Profiling results obtained using DeepSpeed's (Rasley et al., 2020) model profiler, for a batch size of 1 and an image size of $3\times224\times224$. Training recipe refers to the recipes presented in Wightman et al. (2021). ImageNet-1K eval performance obtained from timm (Wightman, 2019) results and our own training. (*): performance evaluated on 256x256 size.

| Model | Parameters | GMACs | GFLOPs | IN-1K performance | Training Recipe |
|---|---|---|---|---|---|
| BN-ResNet-50 | 25.56 M | 4.09 | 8.21 | 78.81 | B |
| GN-ResNet-50 | 25.56 M | 4.09 | 8.24 | 80.06 | A1 |
| SE-ResNet-50 | 28.09 M | 4.09 | 8.22 | 80.26 | B |
| NF-ResNet-50 | 25.56 M | 4.09 | 8.32 | 80.22* | B |
| ANFR-50 (SE) | 28.09 M | 4.09 | 8.32 | 80.4 | B |
| ANFR-50 (ECA) | 25.56 M | 4.09 | 8.32 | 80.61 | B |
| ANFR-50 (CBAM) | 28.07 M | 4.1 | 8.33 | 80.37 | B |

Table 7: Computational demand comparison in a simulated low-resource setting.

| Scenario | Without CUDA | | | With CUDA | |
|---|---|---|---|---|---|
| Metric | Forward | Backward | Total | CPU time | GPU time |
| BN-ResNet-26 | 297ms | 672ms | 969ms | 12ms | 22ms |
| ANFR-26 (ECA) | 353ms | 717ms | 1s 70ms | 9ms | 26ms |

# B    ADDITIONAL RESULTS

## B.1    RESULTS ON FED-ISIC2019 USING FLAMBY HYPER-PARAMETERS

The experimental setup we use for Fed-ISIC2019 in the main paper is an improved version of the example benchmark presented in section 4.1 of Ogier du Terrail et al. (2022), so one might wonder how the compared models perform under the original settings. To answer this we repeat Centralized, FedAvg, and SCAFFOLD training on Fed-ISIC2019 after aligning our hyper-parameters with [11], meaning we reduce local steps to 100 without a scheduler, perform 9 federated rounds, and use pre-computed class weights in the focal loss. Results are presented in Table 8, showing ANFR comprehensively beats competing baselines, with an even wider performance gap compared to our original setting. The overall level of performance, including the gap between centralized and FL training, aligns with the results presented in [11], as we expect. Additionally, SE-ResNet performs better than ANFR in centralized training, but the opposite is true in FL training, further validating our core claims in Section 3 that CA needs Weight Standardization to optimally adjust channel responses in heterogeneous FL. Although these results further support our claims, we believe the optimized version of Fed-ISIC2019 training we provide in the main paper is of use to the community.

Table 8: Results on Fed-ISIC2019 using the original hyper-parameters from FLamby. The gap between ANFR and the baselines is even wider.

|  | BN-ResNet | GN-ResNet | SE-ResNet | NF-ResNet | ANFR (Ours) |
|---|---|---|---|---|---|
| FedAvg | 59.5±0.75 | 55.26±2.96 | 61.92±1.58 | 60.76±0.75 | 65.34±1.29 |
| SCAFFOLD | 57.61±2.78 | 57.62±2.95 | 67.34±0.42 | 57.35±0.73 | 71.07±1.27 |
| Central | 61.26±2.92 | 57.09±1.85 | 73.00±1.09 | 61.28±1.53 | 72.03±1.55 |

## B.2    RESULTS USING RANDOMLY INITIALIZED MODELS

Given the ubiquity and demonstrated utility of ImageNet pre-trained models in FL (Qu et al., 2022; Pieri et al., 2023; Siomos et al., 2024), we use pre-trained models in the main paper. Nevertheless, we conduct additional experiments with FedAvg on CIFAR-10, FedChest and Fed-ISIC2019, using randomly initialized models. Although the results below bolster our claims, we avoided this setting initially as random weight initialization is not representative of the current standard settings adopted by FL practitioners. The only changes made to accommodate the absence of pre-training are to change the optimizer to AdamW and the learning rate to 0.001 for CIFAR-10, and to double the number of local steps for Fed-ISIC2019. Our results in Table 9 show the same trend, of a gap existing between FL and centralized training but being smaller when using pre-trained models. In this setting, too, ANFR is the best performer.

Table 9: Results using randomly initialized models on CIFAR-10, Fed-ISIC2019 and FedChest.

| Dataset | CIFAR-10 | | Fed-ISIC2019 | FedChest | |
|---|---|---|---|---|---|
| Model | FedAvg | Central | FedAvg | FedAvg | Central |
| BN-ResNet | 80.89 | 89.05 | 54.02 | 78.44 | 82.58 |
| GN-ResNet | 78.52 | 86.69 | 54.92 | 73.68 | 80.82 |
| SE-ResNet | 81.19 | 88.65 | 53.2 | 78.79 | 82.16 |
| NF-ResNet | 81.66 | 88.96 | 56.75 | 79.06 | 83.55 |
| ANFR (Ours) | **83.2** | 89.58 | **57.71** | **79.41** | 83.67 |

## B.3    TUNING IN FAVOR OF ANFR IN FED-ISIC2019

As noted in Appendix A.2 which discusses tuning, our hyper-parameters are chosen after tuning the baseline BN-ResNet and not ANFR, meaning the reported improvement in the Tables of the main paper is a conservative floor of the improvement that can be achieved. To illustrate the real impact of our approach, we double the number of local steps in Fed-ISIC2019, keeping all other settings

constant. As seen in Table 10, the performance of ANFR increases by $1.56\%$ compared to Table 1, while its improvement over the best baseline becomes twice as big. While this experimental setting favors ANFR, the performance of BN-ResNet is now lower, so this is not the setting we report in the main paper. The same methodology has been applied for all experimental settings. Despite optimizing for the baselines, ANFR still remains the best option, which greatly bolsters how exciting our results are.

Table 10: Results on Fed-ISIC2019 when doubling the local steps (tuning in favor of ANFR as opposed to BN-ResNet). ANFR performs better than the results in Table \ref{GFL}, but BN-ResNet worse, so this is not the setting used in the main paper

|        | BN-ResNet | GN-ResNet | SE-ResNet | NF-ResNet | ANFR      |
|--------|-----------|-----------|-----------|-----------|-----------|
| FedAvg | 64.52     | 66.16     | 67.55     | 71.76     | **76.34** |

### B.4 PERFORMANCE PLOTS

To gauge convergence, it can be helpful to examine performance plots showing how accuracy progresses throughout federated training. Below we provide four such plots, comparing all models when training from scratch on CIFAR-10 using FedAvg and SCAFFOLD, comparing all models for the experiment in Appendix 10, and a Fed-ISIC run from the top performing model in Table 1, ANFR with SCAFFOLD.

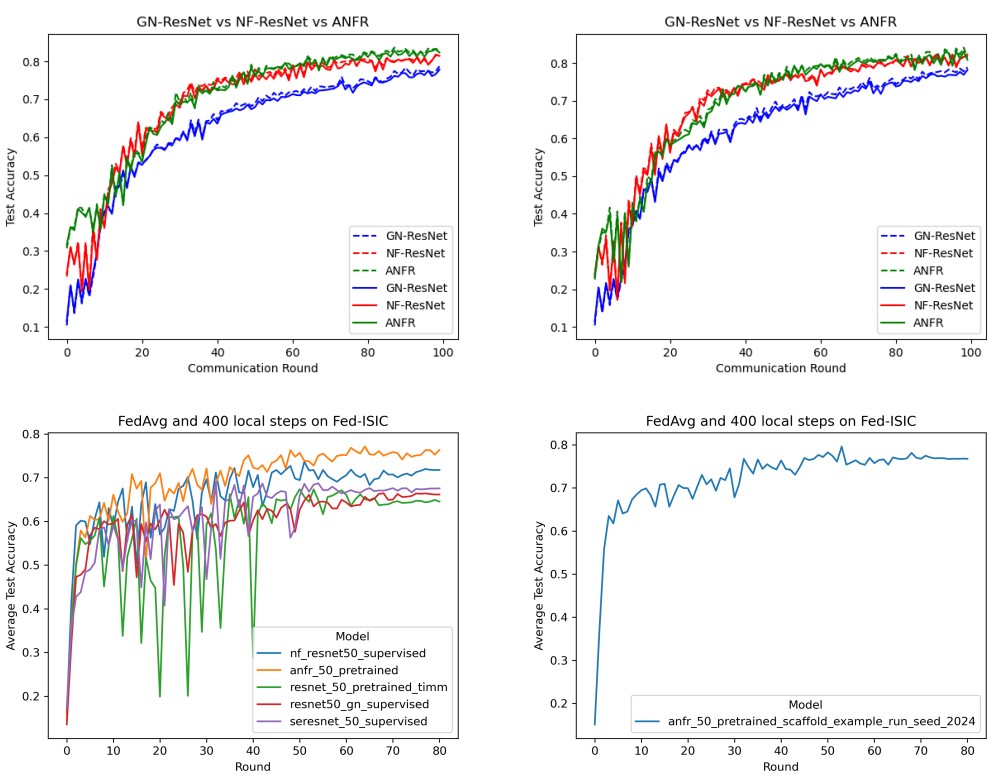

### C TABULAR COMPARISON WITH RELATED WORK

Table 11 present a tabular comparison of ANFR with related work.

Table 11: Comparison of desirable attribute between our study and related work. ○, ◐ , ● symbolize a condition is not met, inconsistently met, and fully met, respectively. ANFR fills a gap by being the first method to simultaneously work in GFL,pFL, and private FL scenarios while being compatible with any aggregation method and offering a robust increase in performance.

| Method | Scenario | | Aggregation Agnostic | Compatible with DP | Performance Increase |
|---|---|---|---|---|---|
| | GFL | pfL | | | |
| FedBN (Li et al., 2021) | ○ | ● | ○ | ○ | ◐ |
| FixBN (Zhong et al., 2024) | ○ | ● | ○ | ○ | ● |
| FBN (Guerraoui et al., 2024) | ○ | ● | ○ | ○ | ● |
| ChannelFed (Zheng et al., 2022) | ○ | ● | ○ | ○ | ● |
| FedWon (Zhuang & Lyu, 2024) | ● | ○ | ○ | ● | ● |
| GN & LN (Wu & He, 2018) (Ba et al., 2016) | ● | ● | ● | ● | ◐ |
| ANFR (Ours) | ● | ● | ● | ● | ● |

## D  FEDCHEST CONSTRUCTION AND ADDITIONAL RESULTS

### D.1  CONSTRUCTION AND HYPER-PARAMETERS

To create **FedChest** we use three large-scale chest X-Ray multi-label datasets: CXR14 (Wang et al., 2017), PadChest (Bustos et al., 2020) and CheXpert (Irvin et al., 2019). To derive a common dataset format for all three, we need to take several pre-processing steps:

1. We remove lateral views where present, keeping only AP/PA views.

2. We discard samples which do not contain at least one of the common diseases, which are: Atelectasis, Cardiomegaly, Consolidation, Edema, Effusion, No Finding, Pneumonia, and Pneumothorax.

3. We remove "duplicates" which, in this context, means samples from the *same patient* that have the *same common labels* but *different non-common labels*.

4. We remove 5% from the edge of each image to avoid blown-out borders and artifacts.

5. We resize the images to 224x224 pixels.

6. We apply contrast-limited histogram equalization (CLAHE) to the images.

In addition to these common steps, some dataset-specific additional pre-processing steps are necessary, namely setting NaN and 'uncertain' labels of CheXpert to 0 (not present), removing corrupted NA rows from CXR14, and removing corrupted images from PadChest.

After pre-processing, CheXpert has twice as many samples as the other datasets, so we further split it into two clients, cxp_young and cxp_old using the median age of the patient population (63 years), leading to a total of 4 clients with train/val/test splits of (given in thousands): 23.7/15/10 for CXR14, 26/15/10 for PadChest, 29.7/15/7.5 for cxp_old and 31/15/7.5 for cxp_young. The task is *multi-label* classification across the 8 common classes.

After tuning, clients perform 20 rounds of 200 local steps with a batch size of 128, the loss function is weighted Binary Cross-Entropy (BCE), and the optimizer Adam with a learning rate of 5e-4,

annealed over training. Data augmentation includes random shifts along both axes, random scaling and rotation, Cutout, and random cropping.

## D.2 Additional FedChest Metrics

Further to the results presented in the main text, since some of the diseases have an unbalanced label distribution, and to also gauge model performance in deployment, we use the validation Receiver Operating Curve (ROC) to find the optimal class thresholds for each client using Youden's Index (Youden, 1950). Having fixed the thresholds to these values, at test-time we measure the average accuracy and F1 score of each model and present the results in Table 12.

Table 12: Classification results on the held-out test set of FedChest obtained by finding the optimal decision threshold on the validation set and using it to binarize predictions.

| Model | BN-ResNet-50 | | GN-ResNet-50 | | SE-ResNet-50 | | NF-ResNet-50 | | ANFR | |
|---|---|---|---|---|---|---|---|---|---|---|
| Metric | Accuracy | F1 | Accuracy | F1 | Accuracy | F1 | Accuracy | F1 | Accuracy | F1 |
| FedAvg | 74.92 | 42.83 | 75.78 | 43.37 | 75.62 | 42.85 | 75.76 | 43.28 | **75.80** | **43.50** |
| FedProx | **74.72** | **42.28** | 73.41 | 41.76 | 74.14 | 41.60 | 74.11 | 41.47 | 74.16 | 41.85 |
| FedAdam | 74.57 | 42.60 | 74.00 | 41.90 | 74.57 | 42.2 | 74.92 | 42.84 | 75.28 | 43.18 |
| SCAFFOLD | 75.55 | 43.34 | 76.38 | 43.85 | 76.18 | 43.65 | 76.27 | 44.02 | 76.41 | 44.07 |
| FedPer | 75.23 | 43.11 | 75.54 | 43.59 | 75.40 | 43.18 | 75.66 | 43.60 | 75.91 | 43.75 |
| FedBN | 75.62 | 43.22 | N/A | N/A | 75.43 | 43.12 | N/A | N/A | N/A | N/A |

## D.3 Batch Size Ablation Study

The absence of a performance gap between BN-ResNet and ANFR on the FedChest dataset when using FedProx (Table 1) motivates us to perform a study ablating the batch size to examine how inconsistent averaging, which is expected to happen for small batch sizes, affects results. We compare BN-ResNet and ANFR, varying the batch size while keeping all other experimental settings unchanged.

Table 13: Batch size ablation study on FedChest using FedProx. Smaller batch sizes more strongly affect BN-ResNet due to inconsistent mini-batch statistics.

| Batch Size | 16 | 32 | 64 | 128 | 256 |
|---|---|---|---|---|---|
| BN-ResNet | 78.67+0.03 | 80.02+0.18 | **81.79+0.18** | **82.14+0.1** | 81.33+0.07 |
| ANFR | **79.20+0.09** | **80.57+0.03** | 81.71+0.16 | **82.14+0.1** | **82.19+0.07** |

Table 13 shows BN-ResNet's performance degrades more than that of ANFR for small batch sizes (16 and 32). ANFR offers significant advantages compared to BN-ResNet for small batch sizes due to the absence of BN. In the main paper hyper-parameters are tuned based on BN-ResNet's performance; as the best BN-ResNet result is achieved with a batch size of 128, this is the one used. While using a large enough batch size can mitigate intra-client variance to a degree, we see that increasing the batch size to 256 reduces BN-ResNet's performance, indicating diminishing returns. This reinforces that increasing batch size is ultimately not a viable solution for addressing BN's limitations in non-IID FL, and new methods, such as ANFR, are necessary to effectively combat statistical heterogeneity.

# E Extended CSI and Attention Weight Analysis

## E.1 Setup details and performance

FL training is performed on the extremely heterogeneous 'split-3' partitioning of CIFAR-10 from Qu et al. (2022), which consists of 5 clients who each have samples only from 2 classes. The training parameters are the same as in Qu et al. (2022) and Section 4.1. All the compared models are pre-trained on ImageNet and have a depth of 50 layers, which results in 16 attention blocks for each

model that uses channel attention. To calculate the channel attention weights and class selectivity index distributions, we use the entire test set of CIFAR-10, passing each class separately through the models to extract class-conditional activations; this is done both before and after FL training.

For channel attention weights, this allows us to store the distributions of weights of each model for each class and channel index. For the CSI, we query the nearest ReLU-activated feature maps before and after each channel attention block—or the equivalent points for the models that do not use such blocks. In `timm` (Wightman, 2019) terminology, we are referring to the output of `act2` as before, and `act3` as after. Comparing before and after distributions for the same network, allows us to isolate the effect of CA in the case of SE-ResNet and ANFR, and observe the baseline effect of moving through the convolutional block on the CSI distribution in BN-ResNet and NF-ResNet. Finally, the histogram of CSI values for each layer is used to draw an approximation of the continuous probability density function for the layer.

### E.2 CSI PLOTS OF ALL LAYERS

From Figure 5, which shows the CSI plots for every layer in the models, we make several observations regarding the class selectivity of each model.

**SE-ResNet.** Before FL training, CA reduces selectivity in all but the last block, in which it normalizes it. This is how CA was designed to function in the centralized setting, aiding feature learning in the first layers and balancing specificity and generalizability in the last layer (Hu et al., 2018). After FL training, the CSI distribution is much more left-skewed in the final block, showcasing how BN, under FL data heterogeneity, prohibits the network's last layers from specializing compared to centralized training.

**NF-ResNet.** Before FL training we see that selectivity generally increases as we move towards the last layers. The CSI distribution of each layer after FL training is very similar to the one before it, indicating that replacing BN with SWS removes the limitation of the last layers to specialize.

**ANFR.** The distributions are generally similar to those of NF-ResNet except for some where CA reduces selectivity, adding to the evidence that part of the role of CA in centralized training is aiding general future learning. After heterogeneous FL training, ANFR inherits NF-ResNets robustness against heterogeneity, and by comparing the last layer of NF-ResNet and ANFR, we note that ANFR overall becomes more specialized.

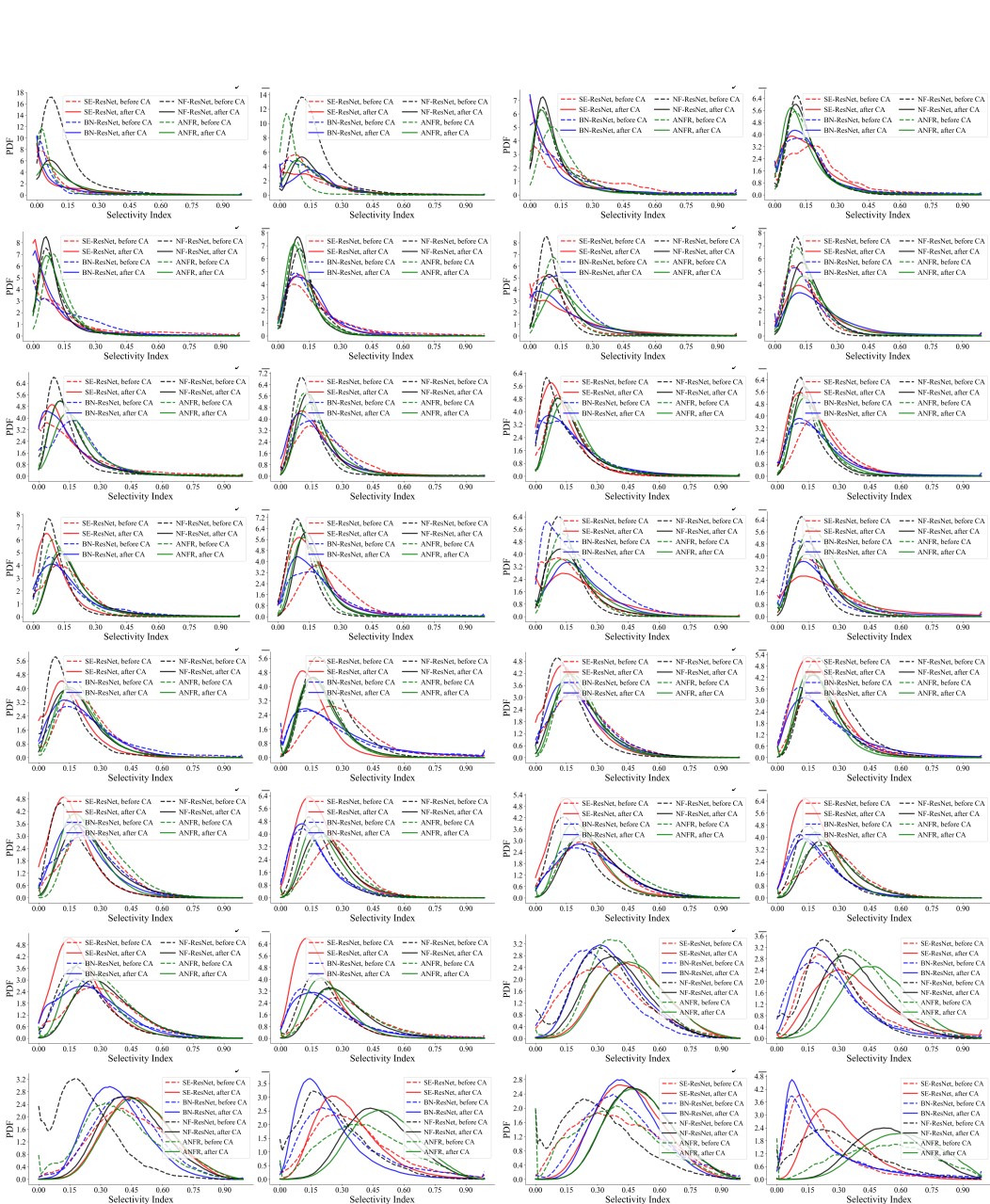

Figure 6: Full CSI results before and after FL training for each layer, moving first across each column then to the next row. In earlier layers CA reduces selectivity, helping the model learn robust features, while in the later ones selectivity is increased to adapt to heterogeneity.

### E.3 CHANNEL ATTENTION PLOTS OF ALL LAYERS

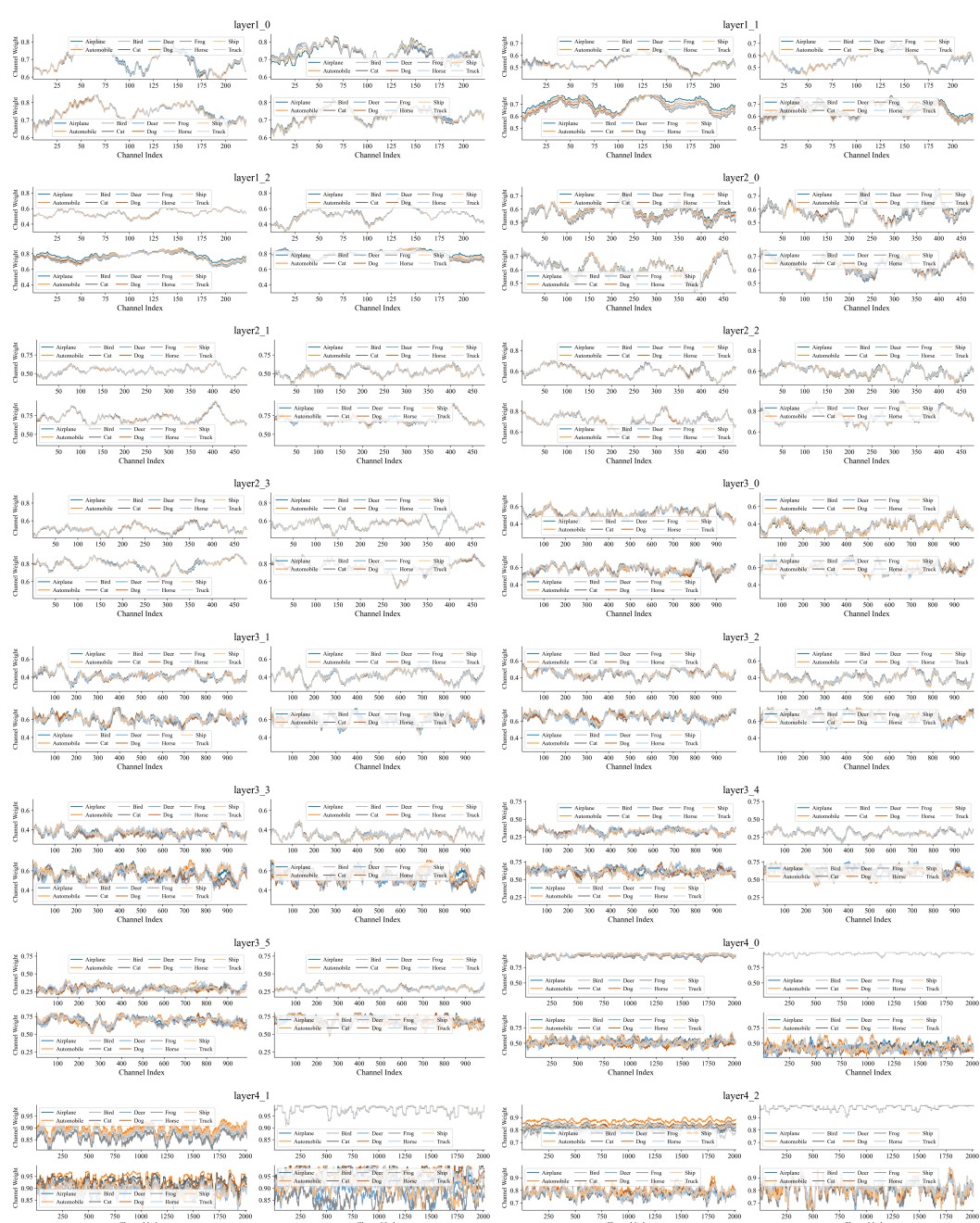

Figure 7: Channel attention weights for every CA module of SE-ResNet and ANFR (top and bottom row of each layer plot, respectively), before and after FL training (left and right). Note the increased variability for ANFR, particularly in the last layer.

