# OpenReview forum: "Addressing Data Heterogeneity In Federated Learning With Adaptive Normalization-Free Feature Recalibration"
_ICLR.cc/2025/Conference — ICLR 2025 Conference Withdrawn Submission_

### Official Review · Reviewer_QmH4 · 2024-10-24

**Soundness:** 2
**Presentation:** 2
**Contribution:** 2
**Rating:** 5
**Confidence:** 4

**Summary:**

This paper presents ANFR, a method designed to address the challenge of statistical heterogeneity in federated learning environments, with a particular focus on the limitations of batch normalization across heterogeneous client datasets. The authors propose a combination of weight standardization and channel attention mechanisms to normalize layer weights and adaptively scale feature maps, aiming to emphasize shared, informative features across clients. The approach is evaluated across various domains, including medical and common computer vision tasks.

**Strengths:**

1. Simple and straightforward approach that can be easily integrated into existing FL frameworks

2. Comprehensive empirical evaluation across multiple datasets

3. Clear experimental setup and ablation studies

**Weaknesses:**

1. The proposed method combines two existing techniques—weight standardization and channel attention—without substantial modifications specific to the federated learning context. The theoretical justification primarily draws from existing literature, lacking novel analysis within the federated learning framework.

2. In Section 3, the connection between feature consistency in different layers and the proposed weight assignment mechanism (A_C_R, A_C_NR) is unclear, undermining the theoretical foundation.

3. The analysis of Figure 2, which aims to demonstrate the method's advantages regarding class selectivity, requires revision. The current presentation shows minimal difference between ANFR and NF-ResNet, contradicting the authors' explanation of significant improvements.

4. (edited and moved the Questions section)

5. (deleted)

**Questions:**

1. Improved organizational structure in Section 3, with clearer problem definition and methodology subsections, would enhance readability. Including detailed pseudo-code would also aid reproducibility.

2. The experiments in Section 4.4 seem disconnected from the core claims. It would be advisable to include content related to this experiment in Section 3.

3. In Table 1, the performance result of FedChest + FedProx + BN-ResNet is identical to that of FedChest + FedProx + ANFR. The authors state that "we employ a large batch size of 128 to maximize the probability that all classes are represented in each batch, following best practices. This choice incidentally biases the setting in favor of BN models since larger batches reduce inconsistent averaging of mini-batch statistics and BN parameters" (lines 427–429). To gain a more detailed understanding, could the authors provide additional results using smaller or larger batch sizes? Additionally, is there further analysis to explain why this result is observed only under the FedProx setting?

---

> ### Author Response · Authors · 2024-11-22
> **First Response to Reviewer QmH4**
>
> We thank the Reviewer for their comments and recognition of some key strengths in our work, such as its simplicity, ease of integration into FL frameworks, and the comprehensiveness of our empirical evaluation. Below, we address the stated weaknesses and suggestions in detail.
>
>
> ## **Novelty (W1)**
>
> We discuss these aspects of ANFR in a separate note above: https://openreview.net/forum?id=uBEl8DMA8K&noteId=ivlFAhRuQt
>
> ## **Connection Between Feature Consistency and Weight Assignment Mechanism (W2)**
> This question refers to lines 190-202 of our submission which we aim to clarify here:
>
> * Convolutional kernels are categorized as consistent/robust against heterogeneity ($C_R$) or inconsistent/non-robust ($C_{NR}$), producing the corresponding activations for each input $A_{C_R}$, $A_{C_{NR}}$, based on their behavior in heterogeneous settings (Figure 1). Consistent activations align well across clients, while inconsistent ones vary significantly.
> * This is merely a virtual distinction we provide solely as a means of explaining the intuition behind why CA works in non-IID FL: CA dynamically recalibrates these channels, enhancing consistent features and suppressing non-consistent features through learnable scaling factors $S$ (Equation 5).
> * As explained in lines 224-252 of our submission, the theoretical foundation of our work lies in deriving and interpreting the input CA receives when Scaled Weight Standardization (SWS) is used, which ensures these weights are derived from stable, weight-based statistics rather than client-specific batch statistics (Equation 8).
>
> We thank the reviewer regarding their feedback on this section and will update the manuscript to better explain this part of section 3 based on the above expounded explanation.
>
> ## **Figure 2 Analysis (W3)**
>
> * Figure 2 compares the Class Selectivity Index (CSI) distributions before FL training (left panel) and after FL training (right panel). The critical insights relate to the post-FL training distributions (right panel), where ANFR shows a distinct rightward shift compared to NF-ResNet.
> * **Concretely, the percentage of neurons with strong class selectivity (CSI > 0.75) is nearly doubled from ~11% to ~21%**, supporting our claim that ANFR better enhances selectivity under FL.
> * Since this analysis is novel and specific to the FL setting, we would also like to highlight it also addresses some of the reviewer’s concerns stated in W1.
> * Lastly, as Figure 2 is part of a mechanistic interpretability study, it provides insights into *why* ANFR works rather than serving as a direct performance comparison, with significant improvements demonstrated through the quantitative results in Section 4.
>
> ## **FedChest Results (W4)**
>
> We are confused by the claim of “contradictory results” in the FedChest experiments. Our results, as presented in Tables 1 and 2, show no inconsistencies; ANFR improves performance across the board. Furthermore, the Reviewer claims we attributed inconsistencies to batch size configurations, which is inaccurate; we make no acknowledgement of inconsistency in the manuscript. If this concern stems from a misinterpretation, we kindly request clarification from the reviewer so that we can address it more thoroughly.
>
> ## **Relevant Baselines (W5)**
>
> We appreciate the reviewer’s acknowledgment of our extensive evaluation and the clarity of our experimental setup. Towards this end we included all relevant baselines. In the scope of our work this encompasses architectural alternatives to batch normalization that have been developed for *centralized learning* and which have been previously explored in FL settings; to the best of our knowledge ours is the first work to design an architectural alternative for the *FL setting*.
>
> If the Reviewer is aware of specific pre-existing “batch normalization alternatives designed specifically for federated learning settings”, we would appreciate the Reviewer pointing them out, as their inclusion and discussion would indeed be a valuable addition to our work.
>
> ## **Placement of Section 4.4 (Q1)**
> Section 4.4 evaluates the performance of different channel attention mechanisms across all datasets introduced in Section 4.1. As such, it logically follows Sections 4.2 and 4.3, which present results using the primary CA mechanism. We believe the ablation study in Section 4.4 strengthens our main findings by showing that careful consideration of the channel attention mechanism used can enhance performance in heterogeneous FL settings even further.
>
> ## **Section 3 Organization and Reproducibility (Q2)**
> We appreciate the suggestion to improve Section 3’s flow. In the revised manuscript, we have added signposts to make its structure clearer. Additionally, we have already committed to open-sourcing our codebase upon acceptance, including pre-trained models and detailed implementation instructions. Our CIFAR-10 codebase is now available for inspection at https://anonymous.4open.science/r/anfr_cifar10/.

---

> > ### Comment · Reviewer_QmH4 · 2024-11-25
> > **Response to the Authors regarding W4**
> >
> > First, I apologize for any misunderstanding regarding W4 caused by the choice of words, such as "contradictory" and "inconsistencies." My intention in W3 was to express curiosity about the lack of a performance gap between FedProx + BN-ResNet and FedProx + ANFR in the FedChest dataset results.   Considering the authors' statements that "BN has been shown to negatively affect performance ..." (lines 47–50) and that "we employ a large batch size of 128 to maximize the probability that all classes are represented in each batch, following best practices. This choice incidentally biases the setting in favor of BN models since larger batches reduce inconsistent averaging of mini-batch statistics and BN parameters" (lines 427–429), these analyses appear directly related to the results I highlighted. However, this result and analysis could be misinterpreted to suggest that simply modifying the batch size could address the limitations of using BN in FL.
> >
> > Thus, I raised this point to request a more detailed explanation of these results (i.e., FedProx + BN-ResNet vs. FedProx + ANFR on the FedChest dataset). Specifically, I would like to see empirical analysis regarding batch size to clarify how reducing inconsistent averaging of mini-batch statistics may have influenced the observed outcomes.
> >
> > By the way, I will clarify W4's statement and move it to the question section to reflect the aforementioned intention.

---

> > > ### Author Response · Authors · 2024-11-27
> > > **Second Response to Reviewer QmH4**
> > >
> > > We thank Reviewer QmH4 for clarifying W4, which has allowed us to better understand their concerns regarding the FedChest results when using FedProx as the aggregation method and our comments on lines 427-429. Below we address these points in detail.
> > >
> > > ---
> > > ## Batch Size Ablation on FedChest using FedProx
> > >
> > > We performed a new ablation study on the effect of batch size on FedChest using FedProx, comparing BN-ResNet and ANFR while keeping all other experimental settings unchanged. Results for batch size 128 are replicated from Table 1 for consistency:
> > >
> > > |Batch Size|16|32|64|128|256|
> > > |-|-|-|-|-|-|
> > > |BN-ResNet|78.67±0.03|80.02±0.18|**81.79+0.18**|**82.14±0.1**|81.33±0.07|
> > > |ANFR|**79.20±0.09**|**80.57±0.03**|81.71±0.16|**82.14±0.1**|**82.19±0.07**|
> > >
> > > ANFR outperforms BN-ResNet for batch sizes 16, 32 and 256, slightly underperforms at 64, and performs equally at 128. The latter is something we believe to be a simple coincidence rather than indicative of any specific phenomenon. **ANFR offers significant advantages compared to BN-ResNet for small batch sizes**, but we reiterate hyper-parameters were tuned based on BN-ResNet's performance; as the best BN-ResNet result is achieved with a batch size of 128, this is the one used in the main paper.
> > >
> > > We hope this new ablation study addresses the Reviewer’s request. While we understand the curiosity regarding the limited performance gap in this specific combination of aggregation method and dataset, we note that FedProx yields the lowest performance on FedChest across all models, even with extensive tuning of the $\mu$ parameter. We believe further investigation into this aspect may not significantly contribute to the broader findings of our work, especially as this behavior is not exhibited in other datasets or aggregation methods.
> > >
> > > ---
> > > ## Regarding Effects of Batch Size in Non-IID FL
> > > One limitation of BN in non-IID FL is the inconsistent averaging of mini-batch statistics. This begins with BN locally capturing mini-batch statistics that might not be representative of the local dataset due to local heterogeneity (intra-client variance), and is further compounded by federated averaging across clients which are non-IID (inter-client variance).
> > >
> > > The sentence at line 728 ("Incidentally...") aims to convey that large batch sizes may help mitigate intra-client inconsistency, as noted in centralized learning literature. However, we agree this sentence is imprecise, and a more accurate statement would be that a *sufficiently* large batch size avoids the pitfalls of small batch sizes, which disproportionately affect BN-enabled networks.
> > >
> > > Referring to the ablation study above, BN-ResNet's performance degrades more significantly than that of ANFR for small batch sizes (16 and 32). However, increasing the batch size to 256 reduces BN-ResNet’s performance, indicating diminishing returns beyond 128. This reinforces that increasing batch size is not a viable solution for addressing BN’s limitations in non-IID FL. This necessitates new methods, such as ANFR, to combat statistical heterogeneity effectively.
> > >
> > > Additionally, “best practices” at line 428 refers specifically to training on multi-label, class-imbalanced datasets like FedChest, not FL best practices. We see how this could be misinterpreted and thank the Reviewer for pointing it out.
> > >
> > > ---
> > > ## Proposed Revisions
> > >
> > > To address these points, we will make the following changes in the revised manuscript
> > >
> > > * Clarify "best practices" at line 428 as "best practices for multi-label, class-imbalanced datasets."
> > > * Remove the sentence "Incidentally..." at line 428 and replace it with a reference to the batch size ablation study in the appendix, providing a more comprehensive analysis including the above comments.
> > > ---
> > > We thank the Reviewer for the time dedicated to our work, and hope this response fully addresses the concerns and requests raised in their reply. Since in their message only W4 was mentioned, we assume that the other concerns had been successfully answered in our initial rebuttal. If this is not the case, we kindly ask the Reviewer to let us know which concerns are still outstanding, so that we can further address them. Otherwise, considering our previous and present rebuttal, the relegation of W4 to a question, and the additional ablation study, we respectfully invite the Reviewer to reconsider their recommendation score for our work.

---

> > > ### Author Response · Authors · 2024-11-29
> > >
> > > We wish to kindly remind the Reviewer we are still awaiting for a response and engagement:
> > >
> > > 1. To our rebuttal regarding every remaining weakness (after the delegation of W4).
> > > 2. To our response and ablation study regarding Q3 (former W4).
> > >
> > > As we believe it to require the biggest time investment on our part, we are especially interested in W5 (lack of comparison to "batch normalization alternatives designed for federated learning settings"). We are still unaware of such previous work, after performing another literature review in response to this concern. If the Reviewer's clarification on which baselines they meant is prompt, we might be able to include some experiments with these methods for the purposes of this discussion.

---

> > > > ### Comment · Reviewer_QmH4 · 2024-12-02
> > > >
> > > > Thank you for the authors' response on Q3. Regarding W5, I clarified the meaning of the baselines based on the other comments from the authors, so I have removed W5.
> > > >
> > > > I have comprehensively reviewed the authors' comments and updated the scores accordingly.

---

> ### Author Response · Authors · 2024-12-02
>
> We thank the Reviewer for removing W5, acknowledging that no other "batch normalization alternatives designed for federated learning settings" exist in the literature.
>
> However, as their updated score remains a 5, effectively a rejection recommendation, we would like to request additional clarifications regarding any remaining concerns:
>
> 1. **Clarification on Novelty (W1)**:
>    Can the Reviewer clarify how they reconcile removing W5, thereby accepting that no other batch normalization alternatives designed for federated learning settings exist, with their continued stance on W1: "lacking novel analysis within the federated learning framework"?
>
>    If no such baselines exist and ours is the first batch normalization alternative designed specifically for federated learning, **it logically follows that the issue of novelty is resolved**. In addition, we reiterate the summary of contributions outlined in our detailed response [here](https://openreview.net/forum?id=uBEl8DMA8K&noteId=ivlFAhRuQt), emphasizing the points that directly address the Reviewer's comments in W1:
>    - ANFR introduces a novel architectural approach to tackle statistical heterogeneity in FL.
>    - Our theoretical analysis (e.g., Equation 8) and interpretability study offer novel insights specific to FL, advancing architectural design research in this field.
>
> ---
>
> 2. **Clarification on W2**:
>    The score of 5 suggests that the Reviewer remains unconvinced regarding W2. However, we provided an extensive rebuttal to this point and would appreciate clarification on any unresolved concerns or gaps in our response that may require further elaboration.
>
> ---
>
> 3. **Clarification on W3**:
>    The Reviewer previously stated that there was "minimal difference between ANFR and NF-ResNet." However, this was disproven in our rebuttal, where we analyzed the raw data and measured a significant increase in high CSI units at ~100%—from ~11% to ~21% of the overall distribution. This is now reflected in the revised manuscript in lines 306-307. If the Reviewer still stands by W3, we kindly ask for clarification on why this detailed evidence does not address their concern.
>
> ---
>
> We remain committed to addressing all concerns constructively and believe these clarifications will facilitate a more transparent and fair assessment of our work.

---

> > ### Comment · Reviewer_QmH4 · 2024-12-02
> >
> > I have added comments on the updated score for clarification, as requested by the authors.
> >
> > Most of my concerns were addressed by the authors' response, except for the part of W1: "The proposed method combines two existing techniques—weight standardization and channel attention—without substantial modifications specific to the federated learning context."
> >
> > After thoroughly reviewing the authors' response to this concern, I remain unconvinced that the proposed method demonstrates sufficient novelty to warrant a score above 5.
> >
> > Therefore, I have updated the score to 5 and decided to keep it unchanged.

---

> > > ### Author Response · Authors · 2024-12-02
> > >
> > > We greatly appreciate that the Reviewer has acknowledged the resolution of 4/5 concerns. We respectfully hope they might similarly revisit their position on W1, and their rejection recommendation, based on the following points for reflection:
> > >
> > > ---
> > >
> > > ### **Consistency in the Review**
> > >
> > > The Reviewer highlighted the fact that ANFR offers a "simple and straightforward approach that can be easily integrated into existing FL frameworks" as **the first Strength** in their Review. We are confused as to how simplicity and straightforwardness, initially recognized as key strengths, are now cited as the sole grounds for rejection. We agree with the Reviewer's initial assessment of these being good attributes of ANFR, so we kindly ask the Reviewer to consider whether adding **unnecessary** complexity, purely for the sake of novelty points, would enhance the impact of this work or simply diminish the clarity and practical value of it.
> > >
> > > ---
> > >
> > > ### **ICLR Guidelines**
> > >
> > > ICLR’s reviewing guidelines emphasize that submissions should provide:
> > >
> > > > "New, relevant, impactful knowledge (incl., empirical, theoretical, for practitioners, etc.)."
> > >
> > > Even if one questions ANFR’s methodological novelty, its contribution of actionable empirical insights for FL practitioners is both substantial and clear, as ANFR offers a ***drop-in*** & ***free-lunch*** replacement for ResNet that improves state-of-the-art performance across 20+ non-IID FL settings. This contribution, supported by theoretical insights (Eq. 8), aligns with the guidelines’ emphasis on advancing impactful knowledge for the community.
> > >
> > > ---
> > >
> > > ### **This Reasoning Could Exclude Impactful Research**
> > >
> > > We also feel it important to note, within the broader context of ML research, this reasoning—that papers should be rejected for not introducing novel architectural components—risks excluding impactful works:
> > >
> > > - As one example among many, the ViT paper (ICLR 2021 oral) created significant impact despite no novel architectural innovations by repurposing transformer architectures for vision tasks.
> > >
> > > - Similarly, ANFR leverages existing components to address FL challenges, focusing on their synergy and applicability to a novel problem domain.
> > >
> > > - If one were to apply the same logic used by the Reviewer here, ViT—despite its groundbreaking insights and wide adoption—should have been rejected for not introducing novel architectural mechanisms.
> > >
> > > ---
> > >
> > > ### **Significance in the Context of FL Research**
> > >
> > > FL research has predominantly focused on novel aggregation methods. ANFR is the first method explicitly designed to address FL challenges at the architectural level, **broadening the scope of FL research** in the process. As such, we believe it would be met with great interest by FL researchers at ICLR.
> > >
> > > ---
> > >
> > > We thank the Reviewer for the discussion over our paper, which triggered new and interesting experiments. Given the above considerations, and that all other concerns have been resolved, we hope the Reviewer will re-evaluate the impact of our work and its strong alignment with ICLR’s criteria and traditions.

---

### Official Review · Reviewer_QRtp · 2024-10-27

**Soundness:** 2
**Presentation:** 2
**Contribution:** 2
**Rating:** 5
**Confidence:** 4

**Summary:**

Statistical heterogeneity among multiple client datasets in federated learning can diminish the system's effectiveness. To address this, this paper introduces Adaptive Normalization-free Feature Recalibration (ANFR), an architecture-level solution combining weight standardization and channel attention. Weight standardization normalizes layer weights, making the system more resilient to client data inconsistencies and irregular averaging. Channel attention produces learnable scaling factors for feature maps, helping suppress those that vary significantly between clients.

By applying weight standardization and channel attention, ANFR can enhance model performance by boosting class selectivity and optimizing channel attention weight distribution, delivering benefits that surpass their individual effects. To improve the privacy guarantee, ANFR with differential privacy achieves the balance between privacy and utility.

**Strengths:**

This paper gives some interesting experimental results, such as performance comparison, pFL aggregation method comparison, and differential privacy training.

**Weaknesses:**

- The proposed approach appears to integrate weight standardization and channel attention in a relatively straightforward manner, with few technical complexities or novel mechanisms to address the combination. Moreover, the experimental results are not very exciting (e.g., the improved performance over FedChest is less than 1%.)
- In Section 4.3, ANFR combines differential privacy to enhance the protection of the model. However, the work does not include a formal analysis or theoretical proof to rigorously substantiate the differential privacy guarantees, causing a lack of clarity on the degree of privacy achieved from the theoretical perspective.
- The experimental results look incremental on CIFAR-10 and FedChest for performance comparison. For example, the improved performance is mostly less than 1%.
- Writing: Regarding the writing style, Section 3 delves directly into technical details without offering an introductory overview. This sudden shift into specifics could benefit from a preliminary summary that provides context and a roadmap for the technical content, enhancing the logical flow of the section.

**Questions:**

1. Why is the improved performance in pFL aggregation not impressive? For example, compared with NF-ResNet, the improvements are from 84.2 to 84.9 and 83.7 to 83.8.

2. Although the authors introduced "... train with strict sample-level privacy guarantees, employing a privacy budget of $\epsilon=1$, followed by training without privacy constraints to illustrate the privacy/utility trade-off of each model...", it remains unclear what the configurations are for $\delta,\sigma$ and other common parameters used in differentially private learning.

3. The paper lacks a formal analysis of differential privacy and an insightful explanation of "enabling strong privacy guarantees without sacrificing performance." Given my understanding, the authors should theoretically prove the improved privacy-utility trade-off.

4. The authors stated the contribution "offers a robust and flexible solution to the challenge of statistical heterogeneity". However, I can not find any theoretical analysis of robustness and privacy relevant to statistical heterogeneity.

---

> ### Author Response · Authors · 2024-11-22
> **Response on Novelty, Contribution, and Complexity of ANFR**
>
> ANFR is the first architecture designed to address the challenge of statistical heterogeneity in FL. As such, it fills in a largely unexplored gap in this space, as the extensive body of existing research into mitigating non-IIDness has been focused on either producing novel aggregation methods or evaluation the behavior of well-established architectures in FL settings. ANFR is complementary to these research streams, unlocking the way for further combatting non-IIDness in FL.
>
> ANFR’s straightforwardness is a **deliberate design goal** and one of its strengths, since it ensures easy integration into existing FL workflows with minimal computational overhead. We believe this goal aligns with the broader community goal of harmonizing FL and centralized research.
>
> While SWS and CA are individually well-established, their synergy, which makes ANFR greater than the sum of its parts, is unique to the FL context as we see by comparing the left and the right panels of figures 2 and 3. Ultimately, we aim to establish ANFR as a versatile and accessible paradigm, where different CA modules can be used depending on the task. Rather than complicating this narrative with a novel CA module, we chose the SE block to highlight how ANFR can function as a drop-in replacement within existing workflows. The investigation in section 4.4 opens a research venue into other attention mechanisms in FL, and in general into architectural blocks that are specifically designed for non-IID FL, since no single mechanism is consistently the best.
>
>
> ## **Summary of Contributions**
>
> - ANFR introduces a novel architectural approach to tackle statistical heterogeneity, a problem previously predominantly addressed through novel aggregation methods.
>
> - ANFR explores and leverages the synergy of SWS and CA (mechanisms previously utilized in other contexts), to achieve consistent improvements in non-IID FL.
>
> - ANFR’s simplicity and flexibility make it a valuable addition to the FL community, capable of integrating seamlessly into existing systems with minimal computational overhead.
>
> - Our theoretical analysis (e.g. Equation 8) and interpretability study offer novel insights specific to FL, advancing architectural design research in this field.
>
> Below we provide more detail on the novel aspects of ANFR in relation to its individual components:
>
> ---
>
> ## **Novelty of ANFR Components in Federated Learning**
>
> While ANFR leverages existing architectural mechanisms, Scaled Weight Standardization (SWS) and channel attention (CA), their application to FL is neither intuitive nor straightforward:
>
> ### **Using SWS to Provide Inputs Unaffected by Non-IIDness to CA**
>
> Weight Standardization was initially proposed as a method to stabilize micro-batch training (batch size 1–2) in centralized settings. Its subsequent development into SWS in normalization-free (NF) architectures demonstrated that SWS could match the performance of BN-enabled networks while avoiding the latter’s batch size limitations. However, these works relied on specific contexts, namely **small batch sizes** and **specialized gradient clipping** to achieve their goals.
>
> **ANFR’s Contribution**: We introduce a novel use of SWS tailored to address statistical heterogeneity in FL. Unlike prior work, we employ SWS to provide channel attention with input statistics unaffected by heterogeneous client distributions, thereby allowing the channel recalibration process to have access to better quality data. Importantly, **ANFR achieves its robust performance increase without requiring small batch sizes, gradient clipping, or task-specific optimizations**. This demonstrates that our approach generalizes well in contexts way beyond those for which SWS was originally designed, a finding we believe is valuable for the community.
>
> ### **Using Channel Attention to Mitigate Heterogeneity**
>
> Channel attention mechanisms, like Squeeze-and-Excitation  (SE) blocks, have proven effective in centralized tasks by adaptively scaling feature maps to emphasize informative features. However, their potential to mitigate statistical heterogeneity in FL has not been explored until now.
>
> **ANFR’s Contribution**: We are the first to leverage CA as an architectural block to directly combat statistical heterogeneity in FL. By recalibrating feature responses at the client level, CA suppresses inconsistent features (those that vary widely across clients) while amplifying shared, robust ones. Our mechanistic study (Section 3) demonstrates that SWS provides stable, distortion-free statistics to CA, enabling it to function effectively under heterogeneous conditions. This interaction is formalized in Equation (8), offering a novel theoretical explanation specific to FL.

---

> ### Author Response · Authors · 2024-11-22
> **First Response to Reviewer QRtp**
>
> We thank the Reviewer for engaging with our work and for their positive comments regarding the performance comparisons, pFL aggregation method evaluation, and DP training experiments our work contains. Below we respond systematically to the concerns raised:
>
> ## Novelty of ANFR & Differentially Private Training:
> We respond to these two issues individually:
>
> * We discuss novelty here: https://openreview.net/forum?id=uBEl8DMA8K&noteId=ivlFAhRuQt
> * We discuss DP considerations here https://openreview.net/forum?id=uBEl8DMA8K&noteId=yM4m9VvFO9
>
> ## Significance of Results and Performance Evaluation (W1, W3 & Q1)
> Our broad experimental evaluation shows that ANFR achieves state of the art results in over 20 settings. We find the consistency with which it outperforms the baselines alone exciting and a highly useful insight to convey to the FL community. Most importantly, as noted in Appendix A.2 which discusses tuning, our hyper-parameters are chosen after tuning the baseline BN-ResNet and **not** ANFR, meaning the reported improvement in our Tables **is a conservative floor of the improvement that can be achieved**.
>
> To illustrate the real impact of our approach, we provide the following new analysis. For Fed-ISIC2019, we double the number of local steps while keeping all other settings constant:
>
> |        | BN-ResNet | GN-ResNet | SE-ResNet | NF-ResNet | ANFR  |
> |--------|-----------|-----------|-----------|-----------|-------|
> | FedAvg | 64.52     | 66.16     | 67.55     | 71.76     | **76.34** |
>
> The performance of ANFR has increased by 1.56%, while its improvement over the best baseline becomes 2x bigger compared to Table 1. While this experimental setting favors ANFR, the performance of BN-ResNet is lower than in Table 1, so this is not the setting we report. The same methodology has been applied for all experimental settings. Despite optimizing for the baselines, ANFR still remains the best option, which we believe greatly bolsters how exciting our results are.
>
> We argue **this not only reaches, but surpasses the threshold set by the reviewing guidelines**, which state "A lack of state-of-the-art results does not by itself constitute grounds for rejection. Submissions bring value to the ICLR community when they convincingly demonstrate new, relevant, impactful knowledge”.
>
> With regards to specific datasets and settings the Reviewer enquires about:
>
> * **FedChest**: FedChest represents a multi-label Chest X-Ray classification task where the reported metric is the average Area Under Receiver Operating Curve (AUROC) across the 7 disease labels and the 4 participating clients. As such, the mean AUROC is a very comprehensive metric, and the reported improvements signify a comprehensive improvement over the baseline.
>
> * **CIFAR-10**: Due to the use of pre-trained models, the baseline accuracy is so high, and the gap between centralized training and FL training so low, that it is to be expected that the performance increase due to the use of ANFR would be proportionate, as is the case. To further explore the effectiveness of our technique, we performed a new analysis (see response to Reviewer [8BWs] (https://openreview.net/forum?id=uBEl8DMA8K&noteId=MCa25WGOpx) where we trained on CIFAR-10 from scratch. In the obtained new results we see that the improvement of ANFR over the baselines is  higher than what we reported in Table 1. We strongly believe these additional results definitively prove that ANFR consistently outperforms the baselines on CIFAR-10.
>
> * **pFL**: For our pFL experiments we report the average performance across the **best**, rather than the last, local models, since this reflects the rational decision each client would make to maximize its utility. Reporting the average of peak performance from each of the local models predictably reduces the gap between FL and centralized training, as well as the individual differences between models. ANFR still remains the most performant method across the board, on two different datasets.
> ## Structure of Section 3
> We appreciate the suggestion to improve Section 3’s flow. In the revised manuscript, we have added signposts to make its structure clearer.
> ## Response to Q4
> We offer a solution to statistical heterogeneity in FL that is:
> - Flexible, as ANFR can be combined with any aggregation method, both in GFL and pFL scenarios, with minimal computational overhead and without the need to re-tune hyper-parameters.
> - Robust, as it achieves state of the art results across a wide variety of experimental settings.
>
> This flexibility and robustness to different scenarios is then empirically proven in our work with extensive experiments. As a solution at the model level, ANFR inherits the robustness and resistance to statistical heterogeneity of any aggregation algorithm it is used with. Moreover, as we show in the theoretical part of our contribution, this is further endowed with a resistance to heterogeneity due to the adaptive recalibration of channel responses in ANFR.

---

> ### Author Response · Authors · 2024-11-22
> **Response Regarding Differential Privacy**
>
> We thank the Reviewer for their inquiry regarding the configurations of differential privacy (DP) parameters, including $\sigma, \delta$ among others.
>
> ## Response to Q2: Differential Privacy Parameters
> ### Regarding $\sigma$
> If the Reviewer refers to the noise multiplier as defined in earlier versions of the Opacus library [9], we direct attention to line 379 of our submission, where we report its value as 1.1. This parameter indirectly controls the variance of the Gaussian noise added to the gradients in the DP-SGD mechanism.
>
> Should the Reviewer instead directly refer to the variance of the Gaussian noise, it is important to note that this would be derived from the noise multiplier as explained previously. Our implementation builds on the DP-SGD framework as implemented in Opacus [9], and the actual variance of the Gaussian noise is calculated via the moments accountant algorithm at the heart of the DP-SGD framework [1].
>
> ### Regarding $\delta$
> Consistent with canonical practices in DP literature [7], and as explicitly stated in lines 378-379, we set $\delta=0.1/D_i$ where $D_i$ is the training dataset size. This ensures a negligible probability of privacy breaches even in extreme cases. For further details, the DP-specific settings are explicitly provided in lines 846–853 of our submission.
>
> We hope this clarification addresses the Reviewer's concerns and demonstrates that our approach relies on standard and widely accepted configurations for DP-SGD.
>
> ## Response to W2, Q3: Formal Analysis of Privacy-Utility Trade-Off
> The Reviewer raises concerns about the lack of a formal analysis of DP guarantees and the privacy-utility trade-off. We respectfully disagree with the assertion that such an analysis is necessary within the scope of our work:
>
> ### Reliance on Established DP-SGD Guarantees:
> Our approach employs the DP-SGD algorithm, whose theoretical underpinnings and privacy guarantees have been rigorously established in seminal works (e.g., Abadi et al. [1]) and have since been widely trusted in both academic research and practical deployments [2-6]. **The DP guarantees of DP-SGD hold independently of the specific model being trained, as they are inherent to its mechanism**. Consequently, reproducing proofs of these guarantees would be redundant and outside the scope of our contribution, which focuses on studying a new model architecture better suited for FL, and that we show to be maintaining its utility characteristics under rigorous DP settings.
>
> ### Demonstrating Privacy-Utility Trade-Off:
> Our paper provides empirical evidence for the privacy-utility trade-off by demonstrating performance under both strict DP constraints (\$epsilon = 1$) and without privacy constraints. These experiments showcase the impact of strong privacy guarantees on utility, supporting our claim of enabling strong privacy without excessive performance degradation. The results, detailed in Section 4.3, align with well-documented trade-offs in the literature, further substantiating our approach.
>
> We trust this response clarifies the soundness of our methodology and demonstrates that our work builds on and respects the foundational principles of DP. We remain committed to improving clarity and will ensure that all relevant settings and details are made as explicit as possible in the revised manuscript by repeating these DP-specific details in Section 4.3 of the revised manuscript.
>
> ## References
>  ---
> [1] Abadi, Martin, et al. "Deep learning with differential privacy." Proceedings of the 2016 ACM SIGSAC conference on computer and communications security. 2016.
>
> [2] Google https://research.google/blog/federated-learning-with-formal-differential-privacy-guarantees/
>
> [3] Training Production Language Models without Memorizing User Data https://arxiv.org/abs/2009.10031
>
> [4] Deep Learning with Label Differential Privacy https://research.google/blog/deep-learning-with-label-differential-privacy
>
> [5] Private Ad Modeling with DP-SGD https://ceur-ws.org/Vol-3556/adkdd23-denison-private-ceur-paper.pdf
>
> [6] An efficient DP-SGD mechanism for large scale NLU models https://www.amazon.science/publications/an-efficient-dp-sgd-mechanism-for-large-scale-nlu-models
> [7] How to DP-fy ML: A Practical Guide to Machine Learning with
> Differential Privacy https://arxiv.org/abs/2303.00654
>
> [8] The Secret Sharer: Evaluating and Testing Unintended Memorization in Neural Networks https://arxiv.org/abs/1802.08232
>
> [9] Opacus: User-friendly differential privacy library in pytorch https://arxiv.org/abs/2109.12298.

---

> > ### Comment · Reviewer_QRtp · 2024-11-27
> >
> > Thank you for the authors' response. I would like to keep my rating.

---

> > > ### Author Response · Authors · 2024-11-28
> > > **Please Adhere to the Guidelines You Agreed To**
> > >
> > > We are deeply disappointed by the Reviewer’s lack of engagement, which undermines the constructive discourse that is central to the review process. The Reviewer’s score of *“marginally below the acceptance threshold”* was based on four stated weaknesses, all of which were addressed during the rebuttal. Moreover, **three of these weaknesses have no scientific merit**:
> > >
> > > 1. **A minor formatting issue**, which was addressed in the revised manuscript by introducing subsections in Section 3.
> > >
> > > 2. **A subjective opinion** that our improvements **over** the state of the art were not *“exciting enough.”* This is **objectively** not grounds for rejection as per the conference’s guidelines, which state:
> > >    > *“A lack of state-of-the-art results does not by itself constitute grounds for rejection.”*
> > >
> > > 3. **Erroneous and invalid claims** about the differential privacy (DP) part of our submission. Specifically:
> > >     - The Reviewer raised a concern about the lack of DP guarantees that was **objectively invalid** and **void of any scientific merit**. As clarified extensively in our rebuttal, this concern contradicts seminal DP literature and reveals a **fundamental misunderstanding** of the field. This raises questions about the Reviewer’s self-assessed confidence score of 4, which suggests familiarity with the topic.
> > >     - The Reviewer incorrectly stated that we did not mention DP hyperparameters, despite this information being **explicitly included** in the manuscript.
> > >
> > > 4. **A claim that our method lacks novelty.**
> > >    While we dedicated significant effort to explaining ANFR’s novelty as the first architecture specifically designed to address FL challenges, we respect the Reviewer’s right to maintain a differing opinion on this subjective matter.
> > >
> > > In summary, **at most one of the four weaknesses remains standing**, yet the Reviewer has opted to keep their score unchanged without further explanation.
> > >
> > > ---
> > >
> > > ## **Failure to Adhere to Guidelines**
> > >
> > > We must raise concerns about the Reviewer’s failure to adhere to the reviewing guidelines, which they agreed to follow upon accepting the invitation to review:
> > >
> > > 1. **Imbalance in strengths and weaknesses**:
> > >    The “Strengths” section of the original review consisted of a single vague sentence, while the “Weaknesses” section went into great detail (regardless of its invalidity) with a frankly jarring difference in tone and grammar between the sections. This violates the guideline to:
> > >    > *“List strong and weak points of the paper. Be as comprehensive as possible.”*
> > >
> > > 2. **Inappropriate use of subjective criteria**:
> > >    The Reviewer used their personal opinion of our results being insufficiently *“exciting”* as grounds for criticism, despite our results surpassing the state of the art across more than 20 settings. This violates the guideline:
> > >    > *“No, a lack of state-of-the-art results does not by itself constitute grounds for rejection.”*
> > >
> > > 3. **Failure to engage during the discussion phase**:
> > >    The Reviewer’s post-rebuttal response consisted of a single sentence maintaining their score, without addressing any of the extensive clarifications, evidence, and new experiments we provided. This violates the guideline:
> > >    > *“It is crucial that you are actively engaged during this phase. Maintain a spirit of openness to changing your initial recommendation.”*
> > >
> > > 4. **Failure to update the final recommendation**:
> > >    Despite our rebuttal conclusively addressing three out of four weaknesses, the Reviewer has not revised their recommendation. This violates the guideline to:
> > >    > *“Update your review, taking into account the new information collected during the discussion phase, and any revisions to the submission.”*
> > >
> > > ---
> > > ## **Request for Reconsideration**
> > >
> > > We urge the Reviewer to consider how they would feel in our position, faced with a review that ignored significant clarifications, and to revise their score accordingly. Given that **three out of four weaknesses lack scientific merit**, we respectfully request that the Reviewer adheres to the guidelines and raises their score to reflect the quality and rigor of our work more fairly.
> > >
> > > ---
> > > ## **Concluding Note**
> > >
> > > The Reviewer’s conduct during the review and rebuttal process has been deleterious to the principles of peer review. **We hope the Area Chair will take into account the Reviewer’s lack of effort, failure to adhere to the guidelines, and refusal to engage constructively during the discussion phase when evaluating our submission.**

---

### Official Review · Reviewer_8BWs · 2024-11-03

**Soundness:** 2
**Presentation:** 2
**Contribution:** 2
**Rating:** 5
**Confidence:** 4

**Summary:**

The paper addresses the challenge of data heterogeneity in federated learning by introducing feature recalibration through two main strategies: weight standardization (which normalizes the weights of layers rather than the activations) and channel attention (which suppresses feature maps inconsistent across clients due to heterogeneity). This approach is agnostic to aggregation methods and performs effectively in standard and personalized federated learning settings.

**Strengths:**

- This paper focuses on an important challenge: statistical heterogeneity in federated learning. Introduces weight standardization and channel attention.
- The experimental evaluation in the paper seems exhaustive, covering a wide range of baseline methods and datasets.

**Weaknesses:**

**Unrealistically high results:** Table 1 illustrates very high and seemingly unrealistic results, which appear to outperform even centralized training outcomes (not directly provided in the paper but evident from the literature).

**Lack of sufficient evidence:**
- The primary concern is with the CIFAR-10 experiments. Achieving 97.42% accuracy is challenging even in the centralized settings, let alone in federated settings. This makes the results highly questionable.
- The reported increase in Fed-ISIC experiment is unusually significant and better than what was presented in the original paper [1]. In the original work, the accuracy achieved using FedAvg, FedProx, and SCAFFOLD does not surpass 60%, and centralized training does not exceed 70%, where [1] uses the EfficientNet architecture with a similar fine-tuning approach.
- The performance of the SCAFFOLD method on the ResNet-50 architecture appears unusually high. In nature, SCAFFOLD is designed to work well in convex settings but is generally expected to struggle in highly non-convex and non-smooth scenarios.

[1] Ogier du Terrail, Jean, et al. "Flamby: Datasets and benchmarks for cross-silo federated learning in realistic healthcare settings." Advances in Neural Information Processing Systems 35 (2022): 5315-5334.

**Questions:**

See weaknesses and provide the following details:

- Include a performance plot for the experiments conducted, showing results for both **Centralized** and **Individual** accuracies.
- Conduct and include an experiment that demonstrates how your method performs when you train from scratch (i.e., without pre-training weights), starting with random weights, and provide those results for comparison.

---

> ### Author Response · Authors · 2024-11-20
> **Rebuttal to Reviewer 8BWs: Summary**
>
> We thank the Reviewer for their time and effort in providing constructive feedback. We appreciate the recognition of the importance of addressing statistical heterogeneity in FL and the acknowledgment of the extensive experimental evaluation in our paper.
>
> We stand by our work, which was conducted with scientific rigor and is fully reproducible. The code for our CIFAR-10 experiments is now provided at https://anonymous.4open.science/r/anfr_cifar10/ for inspection. This anonymous repository contains our experiment logs as well, should the Reviewer wish to examine them or plot the results of specific experiments. We have already committed to open sourcing our entire codebase upon acceptance. We believe the concerns raised stem from a misunderstanding and clarify that our results align with prior literature and established practices.
>
> In summary, our rebuttal:
> 1. Addresses our results using ImageNet pre-trained models on CIFAR-10, proving they are consistent with existing literature, both in the centralized [1,7,8] and the FL setting [1,2,3,5].
> 2. Adds the requested experiments on randomly initialized models and centralized settings, which, like the experiments in the main paper, show that ANFR outperforms baselines. We will include these new results in an appendix of the revised manuscript as well as this response.
> 3. Discusses the reviewer's concern about SCAFFOLD, showing our results are in line with pre-existing reported results [1,4,9,10].
> 4. Clarifies why our Fed-ISIC2019 results are different from the example benchmark in FLamby [11], due to different hyper-paparameters, and how this is something common in related literature [1,9,12], and adds a new experiment where we align our hyper-parameters to those of [11] to facilitate cross-paper comparison.
>
> We hope the clarifications and additional experiments address the Reviewer’s concerns. Our results are consistent with the literature, and the new analyses further validate the soundness of our approach and fully support the claims in our paper. Subsequently, we look forward to further engaging with the Reviewer during the discussion period about the core arguments of our work.
>
> In the following two responses, we address the raised concerns in detail:
>
> ---
> ##### References:
> [1] Pieri, Sara, et al. "Handling data heterogeneity via architectural design for federated visual recognition." Advances in Neural Information Processing Systems 36 (2023): 4115-4136.
>
> [2] Qu, Liangqiong, et al. "Rethinking architecture design for tackling data heterogeneity in federated learning." Proceedings of the IEEE/CVF conference on computer vision and pattern recognition. 2022.
>
> [3] https://github.com/sarapieri/fed_het
>
> [4] https://github.com/NVIDIA/NVFlare/blob/main/examples/advanced/cifar10/cifar10-sim/README.md
>
> [5] Chen, Hong-You, et al. "On the Importance and Applicability of Pre-Training for Federated Learning." The Eleventh International Conference on Learning Representations.
>
> [6] Nguyen, John, et al. "Where to Begin? On the Impact of Pre-Training and Initialization in Federated Learning." The Eleventh International Conference on Learning Representations.
>
> [7] Wightman, Ross, Hugo Touvron, and Herve Jegou. "ResNet strikes back: An improved training procedure in timm." NeurIPS 2021 Workshop on ImageNet: Past, Present, and Future.
>
> [8] https://paperswithcode.com/sota/image-classification-on-cifar-10
>
> [9] Siomos, Vasilis, et al. "ARIA: On the Interaction Between Architectures, Initialization and Aggregation Methods for Federated Visual Classification." 2024 IEEE International Symposium on Biomedical Imaging (ISBI). IEEE, 2024.
>
> [10] Manthe, Matthis, Stefan Duffner, and Carole Lartizien. "Federated brain tumor segmentation: an extensive benchmark." Medical Image Analysis 97 (2024): 103270.
>
> [11] Ogier du Terrail, Jean, et al. "Flamby: Datasets and benchmarks for cross-silo federated learning in realistic healthcare settings." Advances in Neural Information Processing Systems 35 (2022): 5315-5334.
>
> [12] Zhang, Ruipeng, et al. "Grace: A generalized and personalized federated learning method for medical imaging." International Conference on Medical Image Computing and Computer-Assisted Intervention. Cham: Springer Nature Switzerland, 2023.

---

> ### Author Response · Authors · 2024-11-20
> **1/2. CIFAR-10 Results, Federated vs. Centralized Performance, Training from Scratch**
>
> ## **Pre-trained Model Performance on CIFAR-10**
>
> The reported level of test accuracies is common in both FL and centralized literature for models pre-trained on ImageNet:
>
> - Pieri et al. [1] and Qu et al. [2] achieve comparable results in federated settings using pre-trained models on CIFAR-10 with "split-2" partitioning. For our CIFAR-10 experiments, we use the codebase of [1] (which is open-sourced [3]), only modifying the model loading routine to support our own models. Our results align with the performance reported in [1] (Table 2; average of 96.36, maximum of 98.4) and [2] (Table 4; 97.78). In the broader FL literature, where Dirichlet sampling is used to simulate heterogeneity, Chen et al. [5] achieve 90.8% accuracy using a much smaller ResNet-20.
> - In centralized training, 98%+ accuracy is common for ResNet-50 [7] and 99%+ is common in general [8], highlighting that our federated results are not unusual and that they remain below centralized benchmarks. We also see this in [1] (Table 2, looking at the ‘Central’ column).
>
> - The reasons why the gap between centralized and FL training is much smaller for pre-trained models are outside the scope of our work, but Chen et al [5] and Nguyen et al. [6] provide deeper discussion.
>
> ## **Training from Scratch on CIFAR-10, FedChest, Fed-ISIC2019**
>
> Given the ubiquity and demonstrated utility of ImageNet pre-trained models in FL [1, 2, 5, 6, 9], we used pre-trained models in our work. Nevertheless, following the Reviewer’s request, we conduct additional experiments with FedAvg on CIFAR-10, FedChest and Fed-ISIC2019, using randomly initialized models. Although the results below bolster our claims, we avoided this setting initially as random weight initialization is not representative of the current standard settings adopted by FL practitioners.
>
> |              | BN-ResNet | GN-ResNet | SE-ResNet | NF-ResNet | ANFR  |
> |--------------|-----------|-----------|-----------|-----------|-------|
> | CIFAR-10     |           |           |           |           |       |
> | FedAvg       | 80.89     | 78.52     | 81.19     | 81.66     | 83.2  |
> | Central      | 89.05     | 86.69     | 88.65     | 88.96     | 89.58 |
> |              |           |           |           |           |       |
> | Fed-ISIC2019 |           |           |           |           |       |
> | FedAvg       | 54.02     | 54.92     | 53.2      | 56.75     | 57.71 |
> |              |           |           |           |           |       |
> | FedChest     |           |           |           |           |       |
> | FedAvg       | 78.44     | 73.68     | 78.79     | 79.06     | 79.41 |
> | Central      | 82.58     | 80.82     | 82.16     | 83.55     | 83.67 |
>
> The only changes made to accommodate the absence of pre-training are to change the optimizer to AdamW and the learning rate to 0.001 for CIFAR-10, and to double the number of local steps for Fed-ISIC2019. Once again, our results show the same trend, of a gap existing between FL and centralized training but being smaller when using pre-trained models, in line with [5]. **In this setting, too, ANFR is the best performer**.

---

> ### Author Response · Authors · 2024-11-20
> **2/2. Fed-ISIC2019 Reported Results, Alignment with FLamby [11], SCAFFOLD Performance**
>
> ## **Disambiguation of the Fed-ISIC2019 Dataset and the Original Experimental Setup**
>
> We wish to clarify the distinction between the Fed-ISIC2019 dataset introduced in Section 3.2 of [11], and the example benchmark results provided in Section 4 of [11]. We use the dataset, but implement our own choices of local and federated hyper-parameters, as explicitly outlined in Section 4.1 and further detailed in Appendix A.1 of our paper. As stated in  [11] (Section 4, paragraph 1), the datasets are meant to allow for any other experimental setup.
>
> Our setup differs from [11] with a) more local steps and federated rounds, b) a one-cycle scheduler, and c) locally computed focal loss weights. These adjustments are well-motivated and explain the observed improvements in accuracy for all models.
>
> Fed-ISIC2019 is used in this standalone manner in many works with a diverse set of benchmarking hyper-parameters, and many of those report a level of performance similar to ours, such as Pieri et al. [1], Siomos et al. [9] and Zhang et al. [12], the latter of whom show even higher accuracies than ours in some of their settings. Importantly, we would like to reiterate that all models we use employ the same hyper-parameters, which were tuned for the BN-ResNet, before replacing it with each alternative architecture. This means we have provided a more-than-fair benchmark comparison in our paper to advance our core claims, despite a cross-paper comparison of absolute numbers with [11] not being an apples-to-apples one.
>
> ## **New Experiment: Alignment with the benchmark settings of [11]**
>
> To further address the Reviewer’s concerns, we repeat Centralized, FedAvg, and SCAFFOLD training on Fed-ISIC2019, after aligning our hyper-parameters with [11]. Concretely, this means we reduce local steps to 100 without a scheduler, perform 9 federated rounds, and use pre-computed class weights in the focal loss. Results from this experiment are summarized below (mean ± std over 3 seeds):
> | **Method**   | **BN-ResNet** | **GN-ResNet** | **SE-ResNet** | **NF-ResNet** | **ANFR**      |
> |--------------|---------------|---------------|---------------|---------------|---------------|
> | **FedAvg**   | 59.50 ± 0.75  | 55.26 ± 2.96  | 61.92 ± 1.58  | 60.76 ± 0.75  | **65.34 ± 1.29**  |
> | **SCAFFOLD** | 57.61 ± 2.78  | 57.62 ± 2.95  | 67.34 ± 0.42  | 57.35 ± 0.73  | **71.07 ± 1.27**  |
> | **Central**  | 61.26 ± 2.92  | 57.09 ± 1.85  | **73.00 ± 1.09**  | 61.28 ± 1.53  | 72.03 ± 1.55  |
>
> - The overall level of performance, including the gap between centralized and FL training, aligns with the results presented in [11].
> - ANFR comprehensively beats competing baselines, with an even wider performance gap compared to our original setting.
> - SE-ResNet performs better than ANFR in centralized training, but the opposite is true in FL training, further validating our core claims in Section 3 that CA needs Weight Standardization to optimally adjust channel responses in heterogeneous FL.
>
> We will add these results to an Appendix in the revised manuscript. Although these results further support our claims, we believe the optimized version of Fed-ISIC2019 training we provide in the main paper is of use to the community.
>
> ## **Literature Context and Purpose of SCAFFOLD in our Work**
>
> It is our understanding that SCAFFOLD works by reducing the variance in the client updates, thus combatting client drift. While the paper introducing it might theoretically prove better convergence rates only for smooth and convex problems, we would be hesitant to agree this translates to an expectation that in *every* empirical evaluation using complex datasets and deep models SCAFFOLD should struggle. Empirical evaluations, including works such as [1, 9] and Manthe et al. [10], have demonstrated SCAFFOLD can show strong performance in non-convex settings, particularly when paired with pre-trained models.
>
> In our work, SCAFFOLD is included as one of six aggregation methods to display the versatility of ANFR. While we respect the Reviewer's experience or evidence regarding SCAFFOLD's limitations, we make no claim about its universal superiority or its specific performance characteristics. Instead, we aim to clarify that SCAFFOLD outperforming other aggregation methods in specific settings is consistent with existing literature. Where the concerns about SCAFFOLD relate to the transparency of our work and its experimental rigor, we use the NVFLARE implementation of the algorithm; there is an example in the relevant repo comparing different aggregation methods on CIFAR-10 which similarly shows SCAFFOLD to be superior to FedAvg [4].
>
> Moreover, in our Fed-ISIC2019 experiments aligned with [11] (reported above), we observe that SCAFFOLD struggles for certain baselines (e.g., BN-ResNet, NF-ResNet). This highlights the variability of SCAFFOLD’s effectiveness depending on the specific setting, but investigating this phenomenon further is outside the scope of our work.

---

> > ### Comment · Reviewer_8BWs · 2024-11-25
> >
> > I would like to thank the authors for their comments and efforts to address the feedback.
> >
> > Upon inspecting the provided codebase - which I assumed should include the requested plots, as I specifically asked for them - I was told to generate them from the codebase. However, I was unable to locate the specific runs of ANFR because they were not included. **Additionally**, I observed that the authors report **the maximum test accuracy**, which appears to occur during the initial communication rounds. However, the model's performance seems to degrade afterward, suggesting a lack of convergence to a stable or optimal point. I believe this is an important issue worth investigating.
> >
> > While I acknowledge the complexity of the work, the absence of both theoretical guarantees and empirical evidence demonstrating convergence raises concerns.
> >
> > I remain by my original rating of the submission.

---

> > > ### Author Response · Authors · 2024-11-27
> > > **Second Response to Reviewer 8BWs 1/2**
> > >
> > > Before discussing the new claims made in the Reviewer's response, we wish to revisit the original review and its stated weaknesses:
> > >
> > > The Reviewer raised concerns about our CIFAR-10 and Fed-ISIC results compared to the literature. Specifically, each weakness focused on comparisons with prior studies. In our rebuttal:
> > >
> > > 1. We presented extensive bibliographical evidence showing our results align with establised findings for these datasets in both centralized and FL settings.
> > >
> > > 2. At the Reviewer' request, we conducted 25 new experimental runs with randomly initialized models on CIFAR-10, Fed-ISIC, and FedChest, which showed:
> > >
> > >     * Results align with previous FL literature on non-pretrained models.
> > >
> > >     * ANFR continues to outperform baselines in this new setting.
> > >
> > > 3. In response to concerns about Fed-ISIC results exceeding those in the original paper, we performed experiments using the original hyper-parameters, which showed:
> > >
> > >     * Results align with the original paper.
> > >
> > >     * ANFR continues to outperform baselines in this configuration.
> > >
> > > We note the Reviewer’s post-rebuttal response does not appear to explicitly address these points, which formed the basis of initial stated weaknesses. We invite the Reviewer to comment on these aspects to facilitate further constructive discourse on our results. In our next response we will address the newly presented concerns that were not part of the original review in detail:

---

> ### Author Response · Authors · 2024-11-27
> **Second Response to Reviewer 8BWs 2/2: New Claims**
>
> ## **Theoretical Convergence Guarantees**
>
> ANFR operates at the model level and inherits the theoretical convergence guarantees of the aggregation methods it is used with. Convergence guarantees are typically provided at the aggregation method level [13], not the model level, and are thus not applicable in our work.
>
> Instead, in Section 3, we provide theoretical (Eq. 8, lines 242-251) and mechanistic interpretability insights (Figures 2-3, lines 266-306), into why ANFR improves the converge point achieved under these guarantees.
>
> [13] Kairouz, Peter, et al. "Advances and open problems in federated learning." Foundations and trends in machine learning 14.1–2 (2021)
>
> ---
> ## **CIFAR-10 Accuracy Reporting**
> The Reviewer states we report the maximum test accuracy, which is incorrect. For CIFAR-10, we follow the setup from [1, 2], where test accuracy is re-measured whenever validation accuracy improves. This measured test accuracy is then reported, regardless of whether it is lower or higher than the previous test accuracy. This methodology, a form of validation-based early stopping, is valid and distinct from reporting maximum test accuracy. This relates only to the experimental setup for CIFAR-10 (inherited from previously published work); for FedISIC and FedChest we simply report the average test accuracy at every round.
>
> ---
> ## **Updated Code and Experimental Logs**
>
> The Reviewer requested performance plots for centralized and individual accuracies. This request was significant in scope (requiring over 500 plots) and challenging to address within rebuttal constraints. Moreover, the purpose of the plots was unclear, preventing us from tailoring the scope of the request. Based on the Reviewer's response, we infer the request is aimed to assess convergence, and we have provided additional plots addressing this concern.
>
> We identified inconsistencies in the organization and naming of the previously shared logs (e.g. ANFR logs mislabeled as “AFR.”). We apologize for any confusion caused and have corrected these issues, along with expanding the shared codebase and including more experimental logs.
>
> **The updated codebase and logs are available at https://anonymous.4open.science/r/anfr_iclr_updated/, with improvements including**:
>
> * Added codebase for FedChest and FedISIC.
>
> * Provided experiment logs for FedISIC runs, specifically a run with the best reported combination from Table 1 (ANFR+SCAFFOLD), and the runs from the analysis included in our rebuttal to Reviewer QRtp). We have also pre-generated summary plots for these.
>
> * Added support and experimental logs for  CIFAR-10 runs reporting test accuracy every round, identified by the suffix ```ALT_test_every_round```.
>
> * Cleaned up log names and removed irrelevant files.
>
> * Produced plots for all the aforementioned experiments as well as those originally presented. The Fed-ISIC and the comparative CIFAR-10 plots are located inside ```comparative_and_fedisic_plots```, and the ```cifar10_output_logs``` folders each contain png images of plots for validation and test accuracy for easier browsing.
>
> * Included helper scripts for generating accuracy plots.
>
> ---
> ## **Convergence Concerns for ANFR**
>
> To address convergence concerns for CIFAR-10, we modified the validation function to report the test accuracy at every round. We select the three top performing models (namely GN-ResNet, NF-ResNet, and ANFR), and two aggregation methods (FedAvg and SCAFFOLD), and use the new validation function to measure whether ANFR shows the stated lack of convergence. These runs are labeled ```ALT_test_every_round```, and comparative plots are provided in the ```comparative_and_fedisic_plots``` folder.
>
> These experiments confirm that ANFR shows steady performance increases and normal fluctuations. We found no evidence of non-convergence in CIFAR-10, Fed-ISIC or FedChest. For example, in the Fed-ISIC sample run for the top-performing model from Table 1 (ANFR+SCAFFOLD), balanced accuracy converges and stabilizes toward the end of training.
>
> Without specific details from the Reviewer, it is challenging to address this claim comprehensively. However, behavior such as overfitting (test accuracy peaking and then decreasing) is typically due to hyper-parameter choices (e.g. learning rate, optimizer or local steps) rather than the model itself.
>
> ---
> ## **Concluding Remarks**
> We emphasize that these new analyses and visualizations align with the original submission and strengthen the empirical validity of our work.
>
> We hope the inclusion of the FedISIC/FedChest code, expanded experimental logs, and additional plots addresses any remaining concerns, but we are happy to further engage until this is demonstrated in a way the Reviewer finds satisfactory. We look forward to further constructive engagement with the Reviewer and a fair assessment of our work taking into account the information contained in our rebuttals.

---

> > ### Author Response · Authors · 2024-11-28
> > **Minor Correction to FedISIC ANFR+SCAFFOLD plot title needed**
> >
> > Please ignore the title of the plot for the ANFR+SCAFFOLD sample run. It mentions 400 steps, but 200 steps were used for this plot, as in Table 1. We will correct the title at the earliest opportunity.

---

> ### Comment · Reviewer_8BWs · 2024-11-29
>
> I thank the authors for their comments and clarifications. While I appreciate the effort, I remain unconvinced for the following reasons:
>
> 1. **Clarification on test accuracy measurement.** I appreciate the clarification regarding the methodology for measuring test accuracy. I apologize for any misunderstanding caused by my phrasing of “maximum test accuracy.” My intent was to refer to the same setting you described, where test accuracy is re-measured whenever validation accuracy improves. That said, I respectfully disagree with the valuation method. This brings me two key concerns:
>    - How is **validation accuracy** reliably obtained in an FL setting, where client data distributions are inherently non-identical and private?
>    - **Server dataset assumptions.** The paper does not specify that the server possesses a dataset representative of the global distribution. If such an assumption were true, it undermines the core motivation for federated training. Typically, in FL, one cannot assume access to a well-distributed dataset at the server level. This discrepancy should be explicitly addressed.
> 2. **Concerns with missing results in the codebase.** Upon reviewing the provided codebase, I noticed, yet again, that some of the runs were not included, particularly those involving pre-training weights. I took the initiative to generate the missing results, which I have shared here for reference: [https://ibb.co/NW5mz14](https://ibb.co/NW5mz14) [CIFAR10, FedAvg, split-3, same lr, same random seed]. These results reveal that:
>    - The rebuttal does not include essential plots required for proper evaluation.
>    - The proposed method, ANFR, exhibits convergence issues, as shown by the shared results.
>    - In contrast, NF-ResNet shows stable convergence to a point.
>
> At this time, I will maintain my current score. Should the authors wish to provide a response, I remain open to further engagement. However, unless additional clarifications or evidence are provided, I believe this discussion can end here.

---

> > ### Author Response · Authors · 2024-12-01
> >
> > We understand the Reviewer's concerns about the practicality of a centralized validation set in a research paper on non-IID FL. While we respectfully disagree for various reasons, we see no benefit in debating this further here. However, we emphasize three key points:
> >
> > 1. Using a centralized validation/test set, while not the most realistic for deployment scenarios, is an established method for evaluating FL methods in research (e.g., see Section 4.2.1 of Wang et al. [14]).
> > 2. Most of our results are on datasets *without* centralized validation/test sets (FedChest and Fed-ISIC), where ANFR consistently outperforms baselines.
> > 3. This matter is inconsequential to ANFR’s merits and success, as evidenced by the FedChest and Fed-ISIC results.
> >
> > [14] Wang, Jianyu, et al. "A field guide to federated optimization." arXiv preprint arXiv:2107.06917 (2021).
> >
> > ---
> > We are, however, happy to discard the validation set and early stopping, as the Reviewer suggests, and focus on the following:
> >
> > **Why is ANFR overfitting in the plot the Reviewer shared for 'split-3'?**
> >
> > The Reviewer asserts this behavior indicates “convergence issues” with our method, but—as we previously explained—such overfitting behavior is an artifact of specific local hyper-parameter choices. Importantly, this overfitting (and we emphasize it is *not* a convergence issue) is straightforward to avoid:
> >
> > - We disable gradient clipping and increase the batch size from 32 to 64.
> > - Note that this setting is arguably *closer* to “reasonable defaults” than the original setting from [1] (as used in Table 1), as gradient clipping is typically applied only when necessary.
> >
> > **Results with Updated Hyper-Parameters:**
> > 1. ANFR no longer overfits on 'split-3': https://ibb.co/1TYd404.
> > 2. Using the same hyper-parameters for 'split-2,' ANFR outperforms baselines and achieves performance near identical to the early-stopped test accuracy reported in Table 1: https://ibb.co/WVkk1Rf.
> > 3. Logs for all models and aggregation methods using this configuration have been shared in the provided repo.
> >
> > We also note that **NF-ResNet experienced actual convergence issues on 'split-3'** with the original settings, where the loss became `NaN`. While the Reviewer uses “non-convergence” to describe ANFR’s earlier regression, it is important to be precise: ANFR is robust and avoids the instability we observe with NF-ResNet under certain hyper-parameters. Notably, NF-ResNet required gradient clipping for stability in multiple cases (e.g., FedAvg on 'split-3', FedAdam and FedProx in 'split-2'), whereas ANFR did not.
> >
> > ---
> >
> > **Pre-empting Concerns About Stability and Tuning Complexity:**
> >
> > To pre-empt claims that ANFR is tricky to tune or unstable, we provide additional results demonstrating non-overfitting behavior with various hyper-parameter choices:
> >
> > 1. **Disable gradient clipping, reduce learning rate (LR):**
> >    - Reducing LR from 0.03 to 0.01 with batch size 32 eliminates overfitting on 'split-3': https://ibb.co/D5m89XF.
> >    - Evaluating all models with these settings on 'split-2' shows ANFR outperforms baselines again: https://ibb.co/Pg3vsmG.
> >
> > 2. **Change optimizer to AdamW:**
> >    - With AdamW, ANFR does not overfit on 'split-3': https://ibb.co/F0CFyG2.
> >    - While time constraints limited running this configuration for all models, ANFR remains stable and effective.
> >
> > All logs for these experiments are now included in the repo, alongside the ANFR FedAvg and SCAFFOLD logs for the original settings, as previously shared: https://ibb.co/y8fcThg.
> >
> > As a final note, the CIFAR-10 setting we use in Table 1 was one following literature. As we have shown, ANFR maintains superiority even if we deviate from it. If this helps convince the Reviewer, we have no objection to substituting the CIFAR-10 results in Table 1 with any of those presented above, where the validation set is not used, for the camera-ready manuscript.
> >
> > ---
> >
> > **Conclusion:**
> >
> > We have ***conclusively demonstrated that ANFR has no convergence issues on CIFAR-10.*** Convergence on other datasets was never in question. The observed overfitting on 'split-3' was caused by specific hyper-parameters unrelated to our method and is easily avoided with adjustments.
> >
> > The Reviewer has stated a willingness to engage further. For that discussion to proceed on fair and scientific grounds, **we call on the Reviewer to take responsibility for their earlier claims** by:
> >
> > 1. Acknowledging the extensive literature that proves our results are neither highly questionable nor unusual.
> > 2. Acknowledging the extra analyses performed at their request (e.g., random initialization and FLamby-like experiments) demonstrating ANFR’s consistent superiority.
> > 3. Acknowledging the inapplicability of their claim that ANFR requires theoretical convergence guarantees.
> > 4. Acknowledging that ANFR has no “convergence issues” on CIFAR-10, as evidenced here.
> > 5. Acknowledging that we have shared all relevant logs, (even though we find it highly questionable such sharing was required).

---

> ### Author Response · Authors · 2024-11-29
> **Clarifications to the claims made - EDITED**
>
> # EDIT
>
> For the avoidance of any doubt, we will first clarify what happened here/what the Reviewer is showing in that plot. If we as the authors were confused, surely so were other readers:
>
> The Reviewer first implies we are hiding certain results, then claims they “*generate the missing results*”. What actually happened is they have shared a plot of the results **we ourselves generated and shared**, and specifically has plotted the **validation accuracy** in **split-3**, a setting used in our paper only for the interpretability study in Section 3 and *not for the performance comparisons*  in Table 1.
>
> **This is to dispel the implication the Reviewer has uncovered something we hid/did not examine.**
>
> We find frankly baffling the extent to which they mischaracterized *plotting the validation accuracy from some of the shared logs*.
>
> **In the setting actually used in the paper for performance comparisons on CIFAR 10 (i.e. using split-2 instead of split-3) ANFR shows stable performance https://ibb.co/y8fcThg.**
>
> Hence, the argument that ANFR “does not converge” for the dataset used in our submission is erroneous and based on a logical leap.
>
> Having said that, the plot the Reviewer produced using our logs **is** showing that for the hyper-parameters used in the paper, which we did not tune but took from [1], ANFR shows overfitting (and *not* issues of convergence during training) in the extremely heterogeneous partitioning split-3. The question is then, is this overfitting/regression inherent to ANFR? Is ANFR tricky to train stably? The answer is categorically no, as we show below:

---

> ### Comment · Reviewer_8BWs · 2024-12-02
>
> > to generate the missing results (plots)
>
> Thank you for the additional plots you have shared. Dear authors, I would like to clarify a misunderstanding here: when I mentioned generating the missing results (plots), I did not mean that I re-run the experiments but that I generated them. Apologies for any confusion caused.
>
> I appreciate your efforts in addressing my concerns, particularly regarding:
> - the performance plots (please include them in the paper),
> - training from scratch, and
> - Fed-ISIC runs.
>
> But I still disagree with the point the authors made regarding the validation set.
>
> Given these, I have adjusted my rating of the paper accordingly.

---

> ### Author Response · Authors · 2024-12-02
>
> We thank the Reviewer for the acknowledgement of the extra analyses conducted at their request, and the acknowledgement we have provided all the plots they requested.
>
> However, we would also like to ask for a (final) comment on:
> 1. The extensive literature that proves our results are neither highly questionable nor unusual.
> 2. The inapplicability of the claim that ANFR requires theoretical convergence guarantees.
> 3. Whether they still believe ANFR has "convergence issues".
>
> ---
>
> >But I still disagree with the point the authors made regarding the validation set.
>
> We would like to clarify we **removed any disagreement to be had in the context of the submission** as we have re-done the CIFAR-10 experiments **without the validation set**. The logs are in the provided repo, and the revised Table 1 (changes: no gradient clipping, batch size increased to 64, **no validation set**) now looks like this:
>
> | Model | BN-ResNet | SE-ResNet | GN-ResNet | NF | ANFR |
> |---|---|---|---|---|---|
> | FedAvg | 67.39 | 74.75 | 96.73 | 96.62 | **97.45** |
> | FedProx | 86.3 | 94.23 | 95.98 | NaN | **96.63** |
> | FedAdam | 57.43 | 88.93 | 95.32 | NaN | **96.96** |
> | SCAFFOLD | 61.37 | 78.99 | 96.57 | 96.84 | **97.49** |
>
> As mentioned, we have no objection to substituting the CIFAR-10 part Table 1 with the above.
>
> ***This would mean that now no part of the submission uses a centralized validation set, which is what the Reviewer raised a concern with.***
>
> If there remains a disagreement that pertains to something different, kindly clarify.

---

### Official Review · Reviewer_fxJS · 2024-11-03

**Soundness:** 2
**Presentation:** 2
**Contribution:** 2
**Rating:** 6
**Confidence:** 3

**Summary:**

The paper presents an approach for improving collaboration across different clients in FL when the clients are heterogeneous in terms of their data distributions. The contribution of this paper is the method ANFR that combines weight standardization and channel attention to reduce the affect of the data heterogeneity in collaboration. The weight standardization makes the local models' training independent of the batch statistics and channel attention can focus or suppress important and irrelevant features respectively. The usage of CA and its beenfit seems intuitive.

**Strengths:**

1. The authors mention that this approach can be used for existing global and personalized FL algorithms with different aggregation mechanisms.
2. Experiments and ablations - The authors conduct extensive evaluation along with ablations demonstrating the effect of different components.

**Weaknesses:**

1. Adding an extra attention module on channels might not be practical for FL when the devices do not have sufficient compute.

**Questions:**

1. The impact of CA is clear with ablations, can you please provide more insights on how SWS helps the learning? Is it required for CA to work?

---

> ### Author Response · Authors · 2024-11-18
> **Response to Reviewer fxJS, 1/2**
>
> We thank the Reviewer for their thoughtful review and for highlighting the positive aspects of our work, such as the extensive evaluation and ablations, as well as the potential utility of ANFR in diverse FL settings. Below, we address the only stated weakness and answer the reviewer’s question in detail.
>
>
> ### **1. Practicality of ANFR in Limited Compute Scenarios**
> We appreciate the Reviewer’s concern regarding the feasibility of ANFR in resource-constrained settings. We believe that ANFR is practical in these scenarios, as demonstrated in our original submission and further substantiated by a new analysis inspired by the Reviewer’s feedback:
> * **Low computational overhead and performance in cross-device settings**:
>   - As detailed in Appendix A.3 of the submission, the parameter counts, GMACs, and theoretical GFLOPs for ANFR are comparable to those of BN-ResNet, with less than a 10% parameter count difference when using Squeeze-and-Excitation as the Channel Attention mechanism, and a negligible one when using Efficient Channel Attention (ECA). These metrics suggest that the additional computational burden introduced by ANFR ranges from negligible to minimal.
>
>
>    - Moreover, our experiment on CelebA (Section 4.2) demonstrates that ANFR is more performant than baselines in cross-device scenarios.
>
> * **New additional analysis**:
>    - To further address the Reviewer’s concern, we conducted additional experiments simulating low-resource environments. We compared a BN-ResNet-26 with the ANFR version of the same architecture using ECA as the attention mechanism, and a batch size of 32, mirroring a realistic setting for device with limited resources. Average time per iteration of forward + backward pass was measured across 100 iterations using PyTorch’s profiler in two scenarios:
>
>      - Devices without a CUDA-enabled GPU (e.g., smartphones).
>
>       - Devices with CUDA-enabled GPUs (e.g., edge devices such as Nvidia Jetson).
>
>       The results are as follows:
>
>       |   Scenario | Without CUDA |          |         | With CUDA |          |
>       |-----------:|:------------:|:--------:|:-------:|:---------:|:--------:|
>       |     Metric |      Forward | Backward |   Total |  CPU time | GPU time |
>       |  BN-ResNet |        297ms |    672ms |   969ms |      12ms |     22ms |
>       | ANFR (ECA) |        353ms |    717ms | 1s 70ms |       9ms |     26ms |
>
> These results show that ANFR introduces marginal overhead (e.g., ~10% without CUDA, ~15% with GPU) while providing a significant performance advantage. This analysis, now included in the revised Appendix A.3, further supports the practicality of ANFR in constrained settings.

---

> ### Author Response · Authors · 2024-11-18
> **Response to Reviewer fxJS, 2/2**
>
> We continue from the first part of our response, answering the Reviewer's question:
>
> ### **2. Impact of Scaled Weight Standardization (SWS) and Synergy with Channel Attention (CA)**
>
> We thank the Reviewer for highlighting this important aspect of ANFR. This topic is discussed extensively in Section 3 of our paper from two perspectives, the mathematical one in lines 204-251, and a mechanistic interpretability one in lines 253-323. Below, we summarize the key points:
>
> 1. **Does channel attention work when used with activation normalization?**:
>
>    - When employing Activation Normalization (BN, GN, LN), channel attention operates on the output of the activation operation (equation 6). Thus, if this output is distorted by a normalization that is ill -suited to heterogeneous federated learning, as we show is the case for BN ( because mismatched client-specific statistical parameters lead to gradient divergence), GN (because it normalizes within a fixed number of channels which we do not know beforehand and might differ between clients), and LN (because it assumes all channels contribute similarly which is not generally true for CNNs), the attention weights are sub-optimal compared to our method.
>    - As a result, the success of CA with BN will vary; as evidenced in Table 1 of our submission, CA used with activation normalization:
>      - Performs worse than not using CA at all for the FedChest dataset.
>      - Achieves similar performance to not using CA for the Fed-ISIC2019 dataset.
>      - At best, yields smaller performance gains compared to our approach for the CIFAR-10 dataset.
>
>    The exact outcome depends on the task and heterogeneity level, but these results consistently demonstrate that CA is less effective with activation normalization than with SWS.
> 2. **How SWS specifically helps**:
>    - By normalizing convolutional weights instead of activations, SWS avoids issues caused by mismatched batch statistics across clients. This is particularly effective when training from pre-trained checkpoints, as the kernel updates are small and somewhat consistent due to the model already being in a ‘good’ part of the loss landscape. This explains why SWS by itself (“NF-ResNet" in our tables) is also performant, although not to the same degree as ANFR.
>    - Additionally, as shown in Equation (8), SWS provides input statistics directly to CA, enabling better calibration of channel responses. This leads to more robust performance improvements.
>
> 3. **Mechanistic Interpretability approach**:
>
>    - The interplay between SWS and CA is further highlighted in our interpretability study (lines 253–323). While CA improves class selectivity during centralized training, it fails to maintain this improvement in FL when used with activation normalization (Figures 2 and 3, right panels). In contrast, in ANFR the synergy of SWS and CA preserves and enhances class selectivity, even under heterogeneity.
>    - This is a novel result that we believe to be of great interest to the community, as neither channel attention, nor its synergy with weight standardization, specifically in federated learning settings, have been explored for mitigating statistical heterogeneity.
>
> These insights, along with the experimental evidence in our paper, demonstrate that SWS is not only complementary to CA but also essential for its effectiveness in FL settings. We hope this clarifies the Reviewer’s query and further highlights the novel contributions of our work to the field of FL.
>
> ---
> ### **Concluding Remark**
>
> In light of these considerations and the above additional analysis, we hope to have clarified that the concern of impracticality does not apply to ANFR. We kindly ask the Reviewer to consider raising their score to reflect this and their overall positive comments on our work.

---

> > ### Comment · Reviewer_fxJS · 2024-11-22
> > **Response to Authors**
> >
> > Thanks to the authors for the detailed rebuttal. After considering the rebuttal I have updated my score.

---

> ### Author Response · Authors · 2024-11-22
> **Second Response to Reviewer fxJS**
>
> We thank the Reviewer for their message and the updated score. It means a lot for us and our hard work in producing this paper. We gather from the reply we have successfully addressed the Reviewer's concerns, and still appreciate the listed strengths.
>
> We kindly ask if there are other concerns that the Reviewer hasn't expressed, preventing an increase in the attributed score past the point of "marginally above the acceptance threshold".
>
> If this is the case, we would welcome the opportunity to try and address them.

---

> > ### Author Response · Authors · 2024-12-02
> >
> > With the deadline for the end of the discussion phase approaching, we respectfully make a final attempt to seek the Reviewer's engagement, in line with the reviewing guidelines, which emphasize the importance of active participation during this phase.
> >
> > We find it unfortunate that we have not yet received a response from the Reviewer to our request for clarification on what they believe is keeping our paper near the acceptance threshold, as indicated by the score of 6.
> >
> > **If the Reviewer has no remaining concerns, we kindly suggest that a score of 8 would more accurately reflect the strengths of our work, as all previous concerns have been addressed comprehensively.**

---

### Author Response · Authors · 2024-11-28
**List of Changes to the Manuscript**

Following our submission of the revised manuscript, we would like to highlight the changes made as a result of discussions with the Reviewers. Below, we detail these changes and the specific concerns they aim to address:

---

1. **Code Repository and Experimental Logs**:
   - A link to an anonymous repository containing our code, selected experimental logs, and performance plots has been added to the abstract and reproducibility statement.
   - This addresses Reviewer 8BWs' doubts about the veracity of our results and findings and helps resolve Reviewer QmH4's question about reproducibility.

2. **Fed-ISIC2019 Experiments Using FLamby Hyper-Parameters**:
   - Additional experiments using the FLamby hyper-parameters have been added in Appendix B.1.
   - These experiments further address Reviewer 8BWs' concerns regarding our Fed-ISIC2019 results in comparison to those reported in the original FLamby paper.
   - Results show that ANFR outperforms the baselines with an even wider margin under these settings.

3. **Random Initialization Experiments**:
   - At the request of Reviewer 8BWs, we performed additional experiments using randomly initialized models (as opposed to ImageNet-pretrained models) on three datasets: CIFAR-10, Fed-ISIC, and FedChest.
   - These results, presented in Appendix B.2, demonstrate that ANFR continues to outperform the baselines in these settings as well.

4. **Performance Plots for Training Convergence**:
   - Performance plots for training on CIFAR-10 and Fed-ISIC have been added to Appendix B.4.
   - These plots demonstrate the stable performance increase and convergence of ANFR, addressing Reviewer 8BWs' request.

5. **Improved Section 3 Structure**:
   - Sub-sections have been added to Section 3 to improve readability and address formatting concerns raised by Reviewers QmH4 and QRtp.

6. **Fed-ISIC Hyper-Parameter Tuning Favoring ANFR**:
   - A new experiment on Fed-ISIC2019 has been added to Appendix B.3, where we tune hyper-parameters specifically in favor of ANFR (as opposed to the baseline BN-ResNet, as done in the main paper).
   - Results demonstrate that the reported improvements in the main paper represent a conservative floor of the gains achievable with ANFR, addressing Reviewer QRtp's comment that our results were not “too exciting.”

7. **Batch Size Ablation Study for FedChest**:
   - Following discussions with Reviewer QmH4, we removed the comment on the effect of batch size on FedChest results (lines 427–429 in the original submission).
   - This was replaced with a batch size ablation study in Appendix D.3, providing clearer insights into the role of batch size in this scenario.
   - Results show that ANFR consistently outperforms baselines, particularly in small batch size regimes.

8. **Low-Resource Computational Overhead Benchmark**:
   - A computational overhead benchmark in low-resource settings has been added in Appendix A.3 (Table 7), directly addressing Reviewer fxJS's concern regarding practicality.
   - Results show that ANFR introduces minimal overhead, at most 15%, compared to a vanilla ResNet, confirming its suitability for low-resource settings.

9. **Conciseness and Length Adjustments**:
   - Word- and sentence-level edits have been made throughout the manuscript to accommodate the increased content while maintaining compliance with length restrictions.

---

### Note · Authors · 2025-01-27

I have read and agree with the venue's withdrawal policy on behalf of myself and my co-authors.